# An integrated stress response via PKR suppresses HER2+ cancers and improves trastuzumab therapy

Cedric Darini[1], Nour Ghaddar[1,2], Catherine Chabot[1], Gloria Assaker[3,4], Siham Sabri[3,4], Shuo Wang[1], Jothilatha Krishnamoorthy[1], Marguerite Buchanan[1], Adriana Aguilar-Mahecha[1], Bassam Abdulkarim[4,5], Jean Deschenes[6], Jose Torres [3], Josie Ursini-Siegel[1,5], Mark Basik[1,5] & Antonis E. Koromilas[1,5]

Trastuzumab is integral to HER2+ cancer treatment, but its therapeutic index is narrowed by the development of resistance. Phosphorylation of the translation initiation factor eIF2α (eIF2α-P) is the nodal point of the integrated stress response, which promotes survival or death in a context-dependent manner. Here, we show an anti-tumor function of the protein kinase PKR and its substrate eIF2α in a mouse HER2+ breast cancer model. The anti-tumor function depends on the transcription factor ATF4, which upregulates the CDK inhibitor P21[CIP1] and activates JNK1/2. The PKR/eIF2α-P arm is induced by Trastuzumab in sensitive but not resistant HER2+ breast tumors. Also, eIF2α-P stimulation by the phosphatase inhibitor SAL003 substantially increases Trastuzumab potency in resistant HER2+ breast and gastric tumors. Increased eIF2α-P prognosticates a better response of HER2+ metastatic breast cancer patients to Trastuzumab therapy. Hence, the PKR/eIF2α-P arm antagonizes HER2 tumorigenesis whereas its pharmacological stimulation improves the efficacy of Trastuzumab therapy.

[1] Lady Davis Institute for Medical Research, Sir Mortimer B. Davis-Jewish General Hospital, Montreal, QC H3T 1E2, Canada. [2] Division of Experimental Medicine, Department of Medicine, Faculty of Medicine, McGill University, Montreal, QC H4A 3J1, Canada. [3] Department of Pathology, Faculty of Medicine, McGill University, Montreal, QC H3A 2B4, Canada. [4] Research Institute of McGill University Health Centre, Montreal, QC H4A 3J1, Canada. [5] Department of Oncology, Faculty of Medicine, McGill University, Montreal, QC H4A 3T2, Canada. [6] Department of Laboratory Medicine and Pathology, University of Alberta, Edmonton, AB T6G 2H7, Canada. Correspondence and requests for materials should be addressed to A.E.K. (email: antonis.koromilas@mcgill.ca)

Overexpression/amplification of the human epidermal growth factor receptor 2 (HER2)/avian erythroblastosis oncogene B2 (ERBB2) occurs in 15–25% of invasive breast, ovarian and gastric cancers and is associated with poor prognosis[1]. HER2 is a member of the epidermal growth factor receptor (EGFR) family of tyrosine kinases with key roles in cell growth, differentiation, motility and invasion[2]. HER2 predominantly dimerizes with HER3 to induce different downstream pro-survival and tumorigenic pathways including the phosphoinositide 3-kinase (PI3K)-AKT/protein kinase B (PKB) and the mitogen activated protein kinase (MAPK) pathway[2]. Co-expression of HER2 with HER3 is associated with poor prognosis and increased metastasis in breast and ovarian cancers[1,2]. A humanized HER2 monoclonal antibody, known as Trastuzumab or Herceptin has become the standard of care for patients with HER2+ cancers, when it is given alone or in combination with traditional chemotherapy regimens[3]. However, 70% of patients with HER2+ breast cancers demonstrate intrinsic or secondary resistance to Trastuzumab, highlighting the importance of developing new therapies[3].

Translation of mRNAs is intimately involved in cancer through the selective synthesis of proteins involved in proliferation, tumor initiation, progression and metastasis[4]. Different forms of stress, such as DNA damage, oxidative stress, endoplasmic reticulum (ER) stress or stress in the tumor microenvironment (e.g., hypoxia, nutrient deprivation) result in the phosphorylation of the α subunit of the translation initiation factor eIF2 at serine 51 (herein referred to as eIF2α-P)[5]. Phosphorylation of eIF2α is mediated by four kinases, namely, the dsRNA-activated kinase PKR, the PKR-like ER resident kinase (PERK), the general control non-derepressible 2 (GCN2) and heme-regulated inhibitor (HRI), all of which comprise a biological process termed the integrated stress response (ISR)[6]. Increased eIF2α-P results in a global inhibition of mRNA translation initiation, but also facilitates translation of select mRNAs containing small upstream open reading frames (uORFs) in their 5' untranslated region (5' UTR)[7]. These mRNAs encode proteins implicated in the adaptation to stress among which the activating transcription factor 4 (ATF4) plays a prominent role[8]. ISR can promote survival, as well as death in a context dependent manner[5,6]. Among the four ISR arms, the PKR/eIF2α-P arm exhibits anti-tumor effects downstream of interferons, tumor suppressors like the phosphatase and tensin homolog (PTEN) and P53, or after treatments with chemotherapeutic drugs[9–15].

Herein, we show that the PKR/eIF2α-P arm suppresses HER2+ breast cancer growth in mice. We identify the transcription factor ATF4 to be a key mediator of tumor suppression by the PKR/eIF2α-P arm via the upregulation of the cyclin dependent kinase (CDK) inhibitor P21$^{Cip1}$ and activation of the c-Jun-N-terminal kinase 1/2 (JNK1/2). We also demonstrate that the PKR/eIF2α-P arm and its downstream anti-tumor pathways are induced by Trastuzumab in HER2+ breast and gastric tumors. We provide very strong evidence for the therapeutic prospective of the eIF2α-phosphatase inhibitor SAL003 in potentiating the anti-tumor effects of Trastuzumab in HER2+ breast and gastric tumor cells in culture and immune deficient mice. We further show that increased eIF2α-P is an independent positive prognostic marker for time to tumor progression (TTP) and overall survival (OS) of HER2+ breast cancer patients treated with Trastuzumab-based chemotherapy. Our findings demonstrate an anti-tumor function of PKR-eIF2α-P arm in HER2+ cancers. Also, the findings show that increased eIF2α-P is a surrogate biomarker of Trastuzumab treatment, whereas its hyperactivation by pharmacological means sensitizes HER2+ tumors to Trastuzumab treatment.

## Results

**PKR and eIF2α-P suppress NEU breast cancer in mice.** We addressed the implication of PKR and eIF2α-P in HER2-mediated tumorigenesis in a mouse model of HER2+ breast cancer. Specifically, we employed mice with a germline deletion of exon 12 of *pkr* (PKR$^{-/-}$)[16], which are devoid of PKR kinase activity[17]. We also evaluated the role of eIF2α-P in HER2+ breast tumorigenesis using mice with a heterozygous knock-in S51 to alanine (A) mutation of phosphorylated eIF2α (eIF2α$^{S/A}$) because mice with the homozygous knock-in mutation (eIF2α$^{A/A}$) die early after birth[18]. PKR$^{-/-}$, as well as eIF2α$^{S/A}$ mice on FVB/N background were crossed with syngeneic mice expressing an oncogenic variant of rat NEU/HER2 (NEU NDL2-5) from the mouse mammary tumor virus (MMTV) promoter, which was previously shown to induce breast tumors in mice with 100% penetrance[19]. The offspring NEU PKR$^{-/-}$ and NEU eIF2α$^{S/A}$ mice developed mammary gland tumors at a mean time of ~112 or ~120 days, respectively, as compared with ~140 days of tumor formation in NEU mice with intact PKR and eIF2α (wild type, WT) (Fig. 1a). Although there were no appreciable differences in the number of formed tumors in the mammary glands (Fig. 1b), the size of NEU PKR$^{-/-}$ or NEU eIF2α$^{S/A}$ breast tumors was substantially increased compared with wild type NEU tumors (Fig. 1c). Breast tumors from NEU PKR$^{-/-}$ or NEU eIF2α$^{S/A}$ mice contained low levels of eIF2α-P and ATF4 compared with NEU tumors from wild type mice as indicated by immunoblotting (Fig. 1d). We noticed high background levels of eIF2α-P and ATF4 in the breast tumors of wild type NEU mice (Fig. 1d), which was attributed to the expression of the NEU transgene. Specifically, immunoblot analyses of mouse breast tissues indicated that eIF2α-P and ATF4 were decreased in the NEU mice prior to tumor formation and increased in the same mice after tumor formation compared with syngeneic mice lacking NEU (Supplementary Fig. 1). This result indicated a stimulatory effect of the tumor microenvironment on eIF2α-P and ATF4, which could account for the elevated background levels of both proteins in the NEU breast tumors (Fig. 1d). Also, NEU breast tumors impaired for PKR (PKR$^{-/-}$) or eIF2α-P (eIF2α$^{S/A}$) tumors displayed increased proliferation and decreased apoptosis compared with wild type NEU tumors based on immunohistochemistry (IHC) analyses for Ki67 and activated Caspase 3 (Supplementary Fig. 2). These findings supported the anti-tumor effects of PKR and eIF2α-P in mouse NEU breast cancer.

**ATF4 increases P21$^{CIP1}$ and JNK activity in NEU breast tumors.** Because increased NEU tumor growth in PKR$^{-/-}$ and eIF2α$^{S/A}$ mice could be mediated via cell-autonomous, as well as immune regulated mechanisms, we employed a mouse NEU breast tumor cell line, which was derived from MMTV-NEU NDL2-5 mice[19], to impair PKR by the clustered regularly interspaced short palindromic repeats (CRISPR). Two independent clones of depleted PKR (PKR$^{-/-}$) exhibited a substantial increase in proliferation compared with tumor cells with intact PKR (Fig. 2a). Also, PKR$^{-/-}$ tumor cells grew better than PKR$^{+/+}$ tumor cells in immunodeficient SCID mice (Fig. 2b, c). The differences in tumor growth were associated with increased Ki67 and decreased activated Caspase 3 in mouse NEU PKR$^{-/-}$ compared with PKR$^{+/+}$ tumors (Supplementary Fig. 3). This data demonstrated that PKR functions in a cell-autonomous manner to suppress mouse NEU breast tumor growth.

Depletion of PKR in the mouse NEU tumor cells led to decreased eIF2α-P and ATF4 expression (Fig. 2d, e). Also, PKR depletion was associated with decreased expression of the CDK inhibitor p21$^{CIP1}$ and upregulation of the dual specificity protein phosphatase 1 (DUSP1) (Fig. 2d, e). DUSP1 is a type-I cysteine-

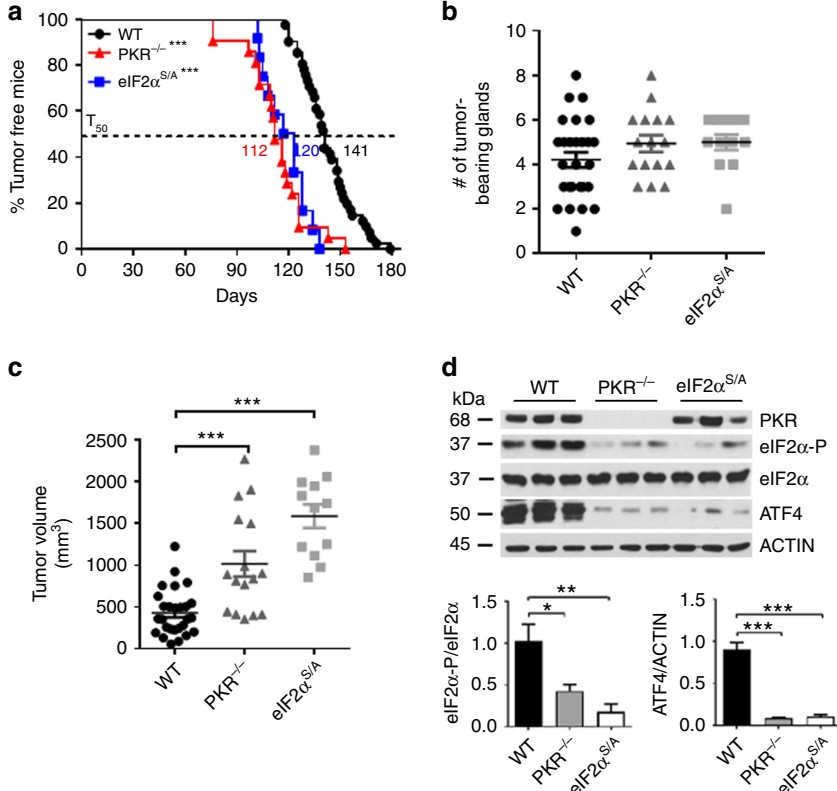

**Fig. 1** Anti-tumor function of PKR and eIF2α-P in mouse NEU breast cancer. **a** NEU wild type (WT; $n = 40$), NEU PKR$^{-/-}$ ($n = 21$) or NEU eIF2α$^{S/A}$ ($n = 12$) mice were monitored up to 180 days for breast tumor formation by palpation. $T_{50}$ represents the time by which 50% of the mouse population develops the first palpable mammary tumor. ***$p < 0.001$. **b**, **c** The total number of tumor-bearing mammary glands, as well as the rate of mammary tumor outgrowth, which is represented by the average tumor volume (mm$^3$), were assessed at 4 weeks after palpation for each of the indicated genotypes ($n > 12$). Data represent mean ± SEM ***$p < 0.001$. **d** Protein extracts of NEU breast tumors from 3 different mice per group were subjected to immunoblotting for the indicated proteins. Graphs indicate the quantification of data from 6 biological replicates. Data represent mean ± SEM. *$p < 0.05$, **$p < 0.01$, ***$p < 0.001$

based protein tyrosine phosphatase, which dephosphorylates the mitogen-activated proteins kinase JNK at both phosphor-Y and phosphor-S residues within the TxY activation motif[20]. In HER2+ breast tumors, DUSP1 displays anti-apoptotic effects by limiting the accumulation of phosphorylated active forms of JNK1/2[21]. Increased DUSP1 expression was inversely proportional to JNK1/2 phosphorylation in PKR$^{-/-}$ compared with PKR$^{+/+}$ tumor cells (Fig. 2d, e). Taken together, the anti-tumor effects of the PKR-eIF2α-P arm are linked to P21$^{CIP1}$ upregulation and increased JNK1/2 phosphorylation in the mouse NEU tumor cells.

Downregulation of ATF4 in mouse NEU tumor cells by two different shRNAs reduced P21$^{CIP1}$ and increased DUSP1, which was accompanied by decreased JNK1/2 phosphorylation (Fig. 3a). Also, ATF4 downregulation was associated with decreased P21$^{CIP1}$ and increased DUSP1 mRNA expression in the mouse breast tumor cells (Fig. 3b, c). This finding was in line with previous reports supporting the transcriptional function of ATF4 in the stimulation of P21$^{CIP1}$ and suppression of DUSP1 in cells subjected to ER stress[22,23]. Moreover, ATF4-deficient cells exhibited increased proliferation rates compared with proficient tumor cells implicating ATF4 in the anti-tumor effects of PKR and eIF2α-P in mouse NEU breast tumors (Fig. 3d).

**DUSP1 inhibition by PKR suppresses NEU breast tumor growth**. Next, we addressed DUSP1's function in mouse NEU tumor cells by targeting its expression with two different shRNAs. Downregulation of DUSP1 increased JNK1/2 phosphorylation

to a greater extent in PKR$^{+/+}$ than PKR$^{-/-}$ tumor cells (Fig. 4a). Inhibition of DUSP1 by the shRNAs did not significantly impact on eIF2α-P indicating that the phosphatase acts downstream of eIF2 in the NEU tumor cells (Fig. 4a). However, downregulated DUSP1 caused a substantial inhibition of NEU PKR$^{-/-}$ tumor growth in SCID mice as opposed to NEU PKR$^{+/+}$ tumor growth, which was marginally affected by impaired DUSP1 (Fig. 4b, c). This data suggested that DUSP1 exhibits a pro-tumorigenic function in mouse NEU tumor cells, which is antagonized by PKR.

DUSP1 downregulation increased JNK1/2 phosphorylation without further inhibiting the tumorigenicity of NEU PKR$^{+/+}$ tumor cells in SCID mice (Fig. 4a–c). This result suggested that the anti-tumor effects of JNK1/2 reach a level above which further activation does not lead to tumor suppression. To assess the effects of JNK1/2 on NEU tumor cells, we treated NEU PKR$^{+/+}$ and PKR$^{-/-}$ tumor cells with the JNK1/2 inhibitor SP600125 in the range of the half-maximal inhibitory concentration (IC$_{50}$ ~50 nM)[24]. Treatments of PKR$^{+/+}$ and PKR$^{-/-}$ tumor cells with SP600125 inhibited JNK1/2 phosphorylation (Supplementary Fig. 4) and increased the proliferation of NEU PKR$^{+/+}$ cells without eliciting a significant effect on NEU PKR$^{-/-}$ cell proliferation (Fig. 4d). Because SP600125 increased the proliferation of NEU PKR$^{+/+}$ cells to the levels identical with those of NEU PKR$^{-/-}$ cells (Fig. 4d), the data suggested that phosphorylated JNK1/2 mediates the anti-proliferative effects of PKR in mouse NEU tumor cells. Collectively, the data showed that ATF4 acts downstream of PKR and eIF2α-P in mouse NEU

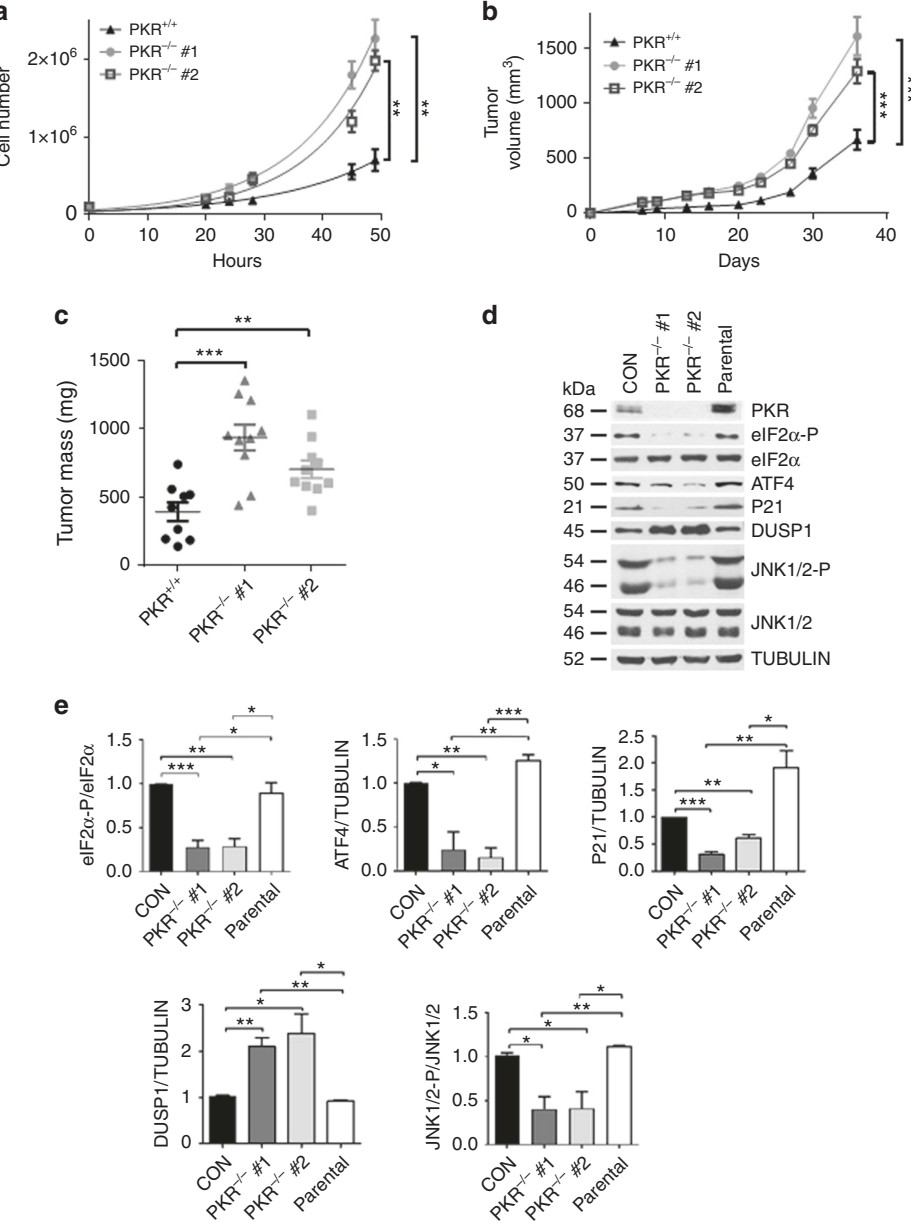

**Fig. 2** Cell-autonomous anti-tumor effects of PKR in NEU breast tumor cells. **a** Proliferation rates of 2 independent clones deleted for PKR by CRISPR (PKR$^{-/-}$) compared with the proliferation of proficient NEU tumor cells (PKR$^{+/+}$). Data is the average of 3 biological replicates. **$p < 0.01$. **b**, **c** NEU PKR$^{+/+}$ and PKR$^{-/-}$ tumor cells ($5 \times 10^5$) were implanted in 5 female SCID mice, two injections per mouse ($n = 2 \times 5 = 10$). Tumor volume (mm$^3$) was monitored for the indicated days (**b**) and tumor mass (mg) was assessed at the experimental endpoint (**c**). Data represent mean ± SEM. **$p < 0.01$; ***$p < 0.001$. **d** Immunoblotting of NEU PKR$^{+/+}$ and PKR$^{-/-}$ tumor cells for the indicated proteins. Protein extracts from the parental mouse NEU tumor cells, which were used for the generation of PKR$^{-/-}$ tumor cells by CRISPR, were included as an additional control. **e** Quantification of blots in d from 3 biological replicates. Data represent mean ± SEM *$p < 0.05$, **$p < 0.01$, ***$p < 0.001$

breast tumor cells to upregulate P21$^{CIP1}$ and increase JNK1/2 phosphorylation via the downregulation of DUSP1 (Fig. 4e).

**Trastuzumab increases eIF2α-P in HER2+ breast tumors.** Downregulation of HER2 by siRNAs increased the autophosphorylation of PKR at T446, which is essential for its activation[25], and increased eIF2α-P in human HER2+ BT474 tumor cells (Supplementary Fig. 5a). Treatment with Trastuzumab increased PKR T446 phosphorylation and eIF2α-P in BT474 cells, which were sensitive (S) to Trastuzumab (Supplementary Fig. 5b). BT474$^S$ cells responded to Trastuzumab by upregulating ATF4 and its downstream targets, namely, P21$^{CIP1}$ and phosphorylated

JNK1/2 via the downregulation of DUSP1 (Supplementary Fig. 5b). On the other hand, Trastuzumab-resistant BT474$^R$ cells exhibited increased background levels of PKR T446 phosphorylation and eIF2α-P compared with BT474$^S$ cells due to continuous treatment with Trastuzumab for the establishment of resistant cells (Supplementary Fig. 5b)[26,27]. However, Trastuzumab did not further increase PKR T446 phosphorylation, eIF2α-P, P21$^{CIP1}$, and JNK/1/2 phosphorylation in BT474$^R$ cells suggesting that the sensitivity of HER2+ breast tumor cells to Trastuzumab is characterized by the induction of the anti-tumor PKR/eIF2α-P arm (Supplementary Fig. 5b).

We also detected eIF2α-P in HER2+ breast tumors from patients treated with Trastuzumab, who developed resistance to

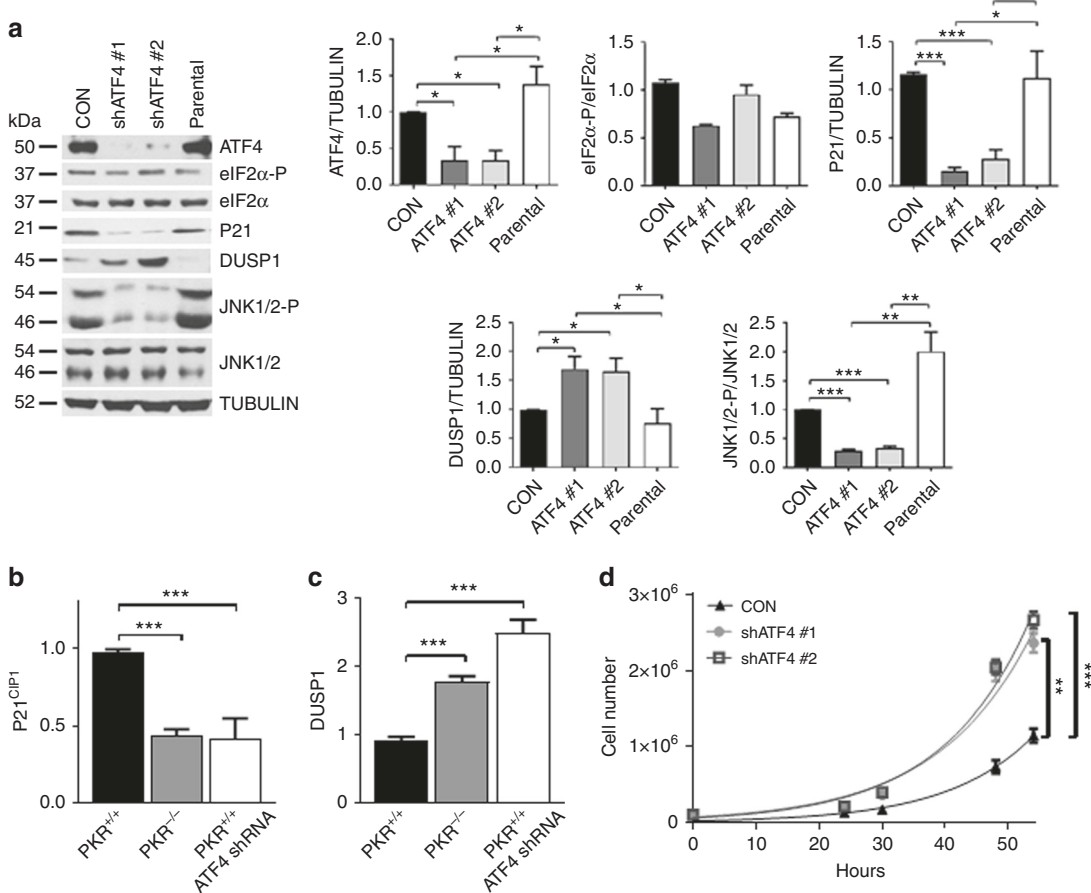

**Fig. 3** ATF4 upregulates P21CIP1 and JNK phosphorylation in mouse NEU tumor cells. **a** Immunoblot analyses of NEU tumor cells that are proficient (CON, cells infected with insert-less lentiviruses) or deficient for ATF4 by the lentivirus-mediated expression of 2 different shRNAs. An equal amount of protein of the parental mouse NEU tumor cells were included as an additional control. Graphs show quantification of blots from 3 biological replicates. **b**, **c** Detection of P21CIP1 or DUSP1 mRNA in NEU PKR[+/+], NEU PKR[-/-] and NEU PKR[+/+] cells expressing ATF4 shRNA by qPCR. Total RNA from cells expressing the 2 different ATF4 shRNAs separately was combined for the qPCR assays. GAPDH and actin mRNAs were used as internal controls. Data were obtained from the analysis of 3 biological replicates and represent mean ± SEM. ***$p < 0.001$. **d** Proliferation rates of NEU PKR[+/+] or NEU PKR[+/+] expressing 2 different ATF4 shRNAs. Data represent the average of 3 biological replicates and represent ± SEM. **$p < 0.01$ ***$p < 0.001$

treatment. IHC staining of the HER2+ breast cancer biopsies indicated that eIF2α-P was more highly increased in tumors after Trastuzumab treatment compared with tumors from the same patients before the treatment (Supplementary Fig. 6). This data agreed to increased background eIF2α-P in BT474[R] compared with BT474[S] cells (Supplementary Fig. 5b), which suggested that increased eIF2α-P is an effect of tumor resistance to Trastuzumab therapy. Detection of activated PKR in the HER2+ breast tumor samples was not possible because the T446 phospho-specific antibody was not suitable for IHC. These data indicated that Trastuzumab therapy increases eIF2α-P in patient-derived HER2+ breast tumors.

**eIF2α-phosphatase inhibition improves Trastuzumab's efficacy.** Next, we tested the effects of the pharmacological stimulation of eIF2α-P in single and combined treatments with Trastuzumab. Specifically, HER2+ breast tumor cells were treated with SAL003, which is a potent derivative of the eIF2α-specific phosphatase inhibitor Salubrinal[28]. Treatment with SAL003 increased eIF2α-P, ATF4, P21CIP1, and JNK1/2 phosphorylation, as well as decreased DUSP1 in BT474[S] and BT474[R] cells (Fig. 5a). These signaling effects of SAL003 were further enhanced by co-treatment of BT474[S] and BT474[R] cells with Trastuzumab

(Fig. 5a). Although HER2 expression was higher in BT474[R] than BT474[S] cells, treatment with SAL003 did not influence HER2 levels in the two cell types indicating that eIF2α-P is not connected to regulation of HER2 expression in the breast tumor cells (Fig. 5a).

Also, SAL003 exhibited a strong inhibitory effect on the colony forming efficacy of BT474[S] and BT474[R] cells, which was further enhanced by co-treatment with Trastuzumab (Fig. 5b). As with BT474 cells, SAL003 exhibited a stronger inhibitory effect than Trastuzumab on the colony forming efficacy of patient-derived HER2+ breast tumor cells and significantly potentiated the inhibitory effects of Trastuzumab on these cells in combined treatments (Fig. 5c). These data revealed the strong antiproliferative properties of SAL003 in HER2+ breast tumor cells, which are further enhanced in co-treatments with Trastuzumab.

We further tested the effects of SAL003 and Trastuzumab on the growth of patient-derived xenografts (PDXs) established from a patient with HER2+ gastric cancer with resistance to Trastuzumab. Growth of the HER2+ gastric cancer PDX in NOD/Shi-scid/IL-2Rγ[null] (NOG) mice was partially sensitive to single treatments with either SAL003 or Trastuzumab, but it became highly susceptible to combined treatments (Fig. 6a). Immunoblot analyses of tumors from NOG mice indicated that SAL003 increased eIF2α-P, ATF4, and P21CIP1 and decreased

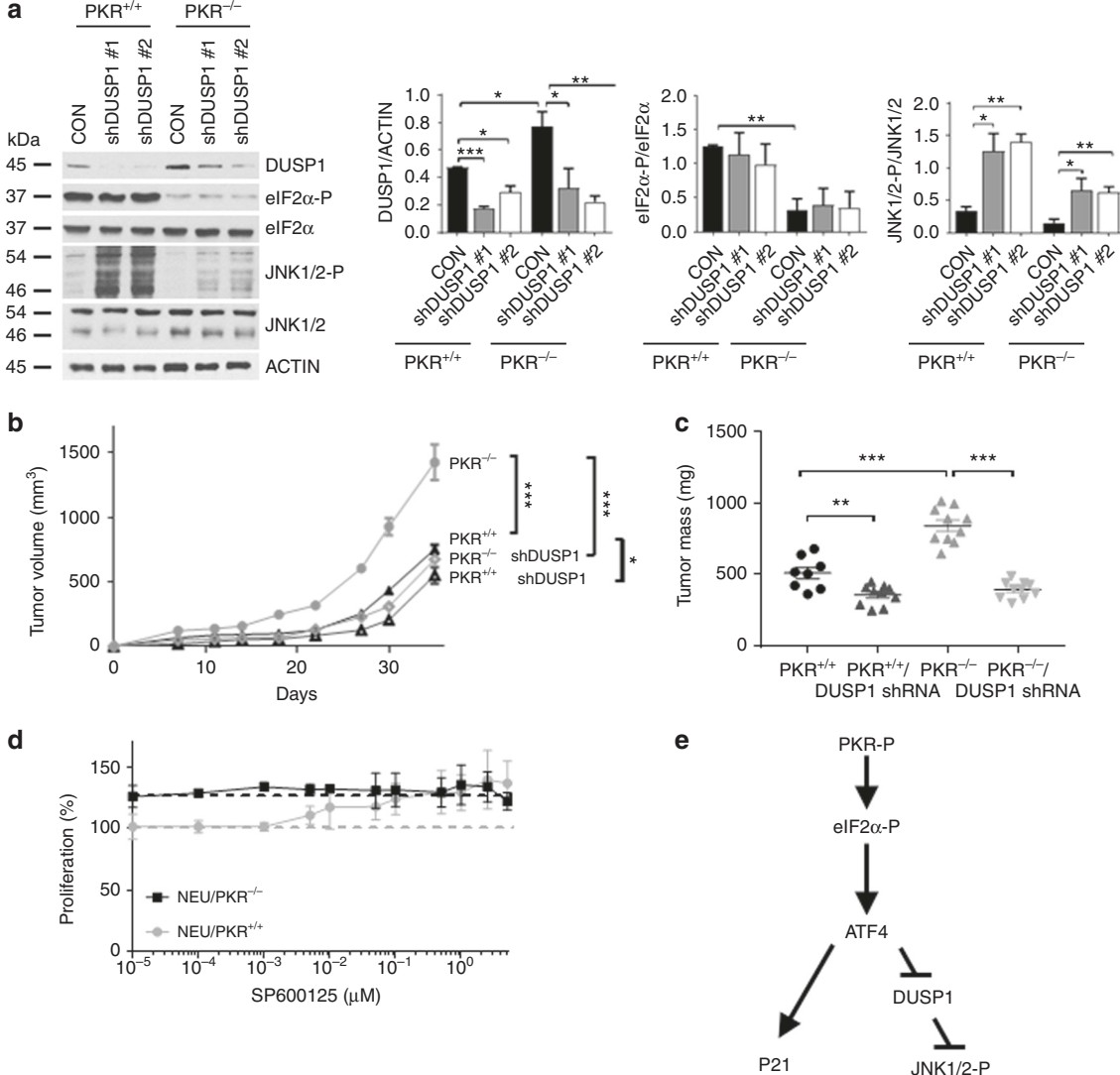

**Fig. 4** PKR-dependent DUSP1 inhibition contributes to mouse NEU tumor suppression. **a** Immunoblot analyses of NEU PKR$^{+/+}$ and PKR$^{-/-}$ tumor cells lacking (CON, cells infected with insert-less lentiviruses) or expressing 2 different DUSP1 shRNAs by lentivirus infection. The graphs represent the quantification of blots from 3 biological replicates and are shown as mean ± SEM *$p < 0.05$; **$p < 0.01$; ***$p < 0.001$. **b** Tumor growth of proficient, as well as DUSP1-deficient (shRNA) NEU PKR$^{+/+}$ and PKR$^{-/-}$ cells in 5 SCID mice. Each mouse received two injections of $5 \times 10^5$ cells ($n = 2 \times 5 = 10$). Transplantation of mice with DUSP1-deficient cells was performed with a mix of an equal number of cells ($2.5 \times 10^5$) expressing DUSP1 shRNA 1 or 2. **c** Graphs represent the mass (mg) of proficient and DUSP1-deficient tumors in SCID mice at 35 days post transplantation. **b**, **c** Data represent mean ± SEM. *$p < 0.05$; **$p < 0.01$; ***$p < 0.001$. **d** Proliferation rates of NEU PKR$^{+/+}$ or NEU PKR$^{-/-}$ cells measured by Sulforhodamine B in the presence of the indicated concentrations of the JNK1/2 inhibitor SP600125 for 3 days. Data represent the average of 3 biological replicates. **e** Schematic representation of the signaling properties of PKR and eIF2α-P. ATF4 functions downstream of eIF2α-P to increase P21$^{CIP1}$ and decrease DUSP1 leading to stimulation of JNK1/2 activity and inhibition of proliferation of mouse NEU tumor cells

DUSP1, which was associated with increased JNK1/2 phosphorylation (Supplementary Fig. 7). Also, combined treatments of NOG mice with SAL003 and Trastuzumab further increased the anti-tumor pathways downstream of eIF2α-P (Supplementary Fig. 7). Evaluation of tumor size at day 44 after the initiation of the anti-tumor treatments indicated a synergistic effect of SAL003 and Trastuzumab in the inhibition of tumor growth in mice (Fig. 6b). Also, vehicle-treated mice died within 44 days, whereas mice treated with either SAL003 or Trastuzumab survived for 16 days longer (Fig. 6c). The life span of 3 out of 5 mice subjected to combined treatment with SAL003 and Trastuzumab was further increased by 13 days, whereas 2 out of 5 mice remained alive for at least 110 days (Fig. 6c). Taken together, the findings with SAL003 suggested that stimulation of eIF2α-P

has the capacity to potentiate the anti-tumor effects of Trastuzumab in HER2+ cancers that are either sensitive or resistant to Trastuzumab therapy (Fig. 6d).

**eIF2α-P is a potential biomarker of Trastuzumab efficacy**. We also analyzed gene expression profiles of HER2+ breast tumors from public data bases (www.kmplot.com) and found that high PKR mRNA levels are associated with increased relapse-free survival (RFS) and increased distant metastasis free survival (DMFS) of HER2+ breast tumor patients, who underwent neoadjuvant chemotherapies (Fig. 7a). We further addressed the prognostic value of eIF2α-P from the analyses of tissue microarrays (TMAs) derived from patients with HER2+ metastatic breast cancer treated with Trastuzumab-based chemotherapy[29].

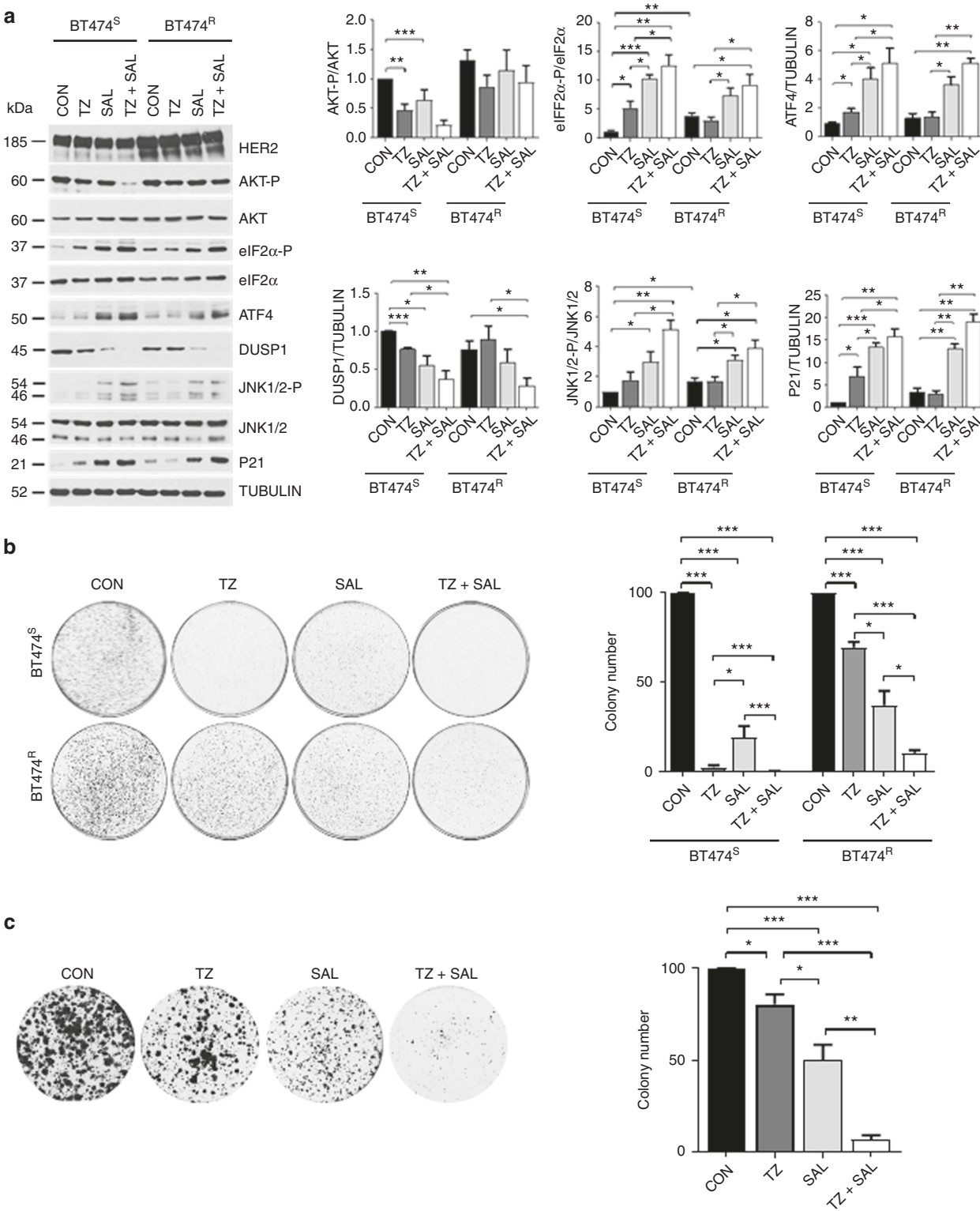

**Fig. 5** The eIF2α-phosphatase inhibitor SAL003 potentiates Trastuzumab's anti-proliferative effects. **a** Immunoblot analyses of protein extracts from sensitive (S) or resistant (R) to Trastuzumab BT474 cells treated with 21 μg/ml Trastuzumab, 10 μM SAL003 or both for 3 h. Graphs represent the quantification of data from 3 biological replicates and are shown as mean + SEM *$p < 0.05$; **$p < 0.01$; ***$p < 0.001$. **b, c** Evaluation of the colony forming efficacy of Trastuzumab-sensitive (S) or resistant (R) BT474 cells (**b**), as well as of Trastuzumab-resistant HER2+ breast cancer BM-132 cells (**c**) after treatment with 0.02% v/v DMSO (CON), 21 μg/ml Trastuzumab (TZ), 5 μM SAL003 or both drugs at the same concentrations as the single treatments (TZ + SAL). The graphs show the % inhibition of the colony forming efficacy of the cells after normalization to CON cells. The data were obtained from 3 biological replicates and are shown as mean ± SEM *$p < 0.05$; **$p < 0.01$; ***$p < 0.001$

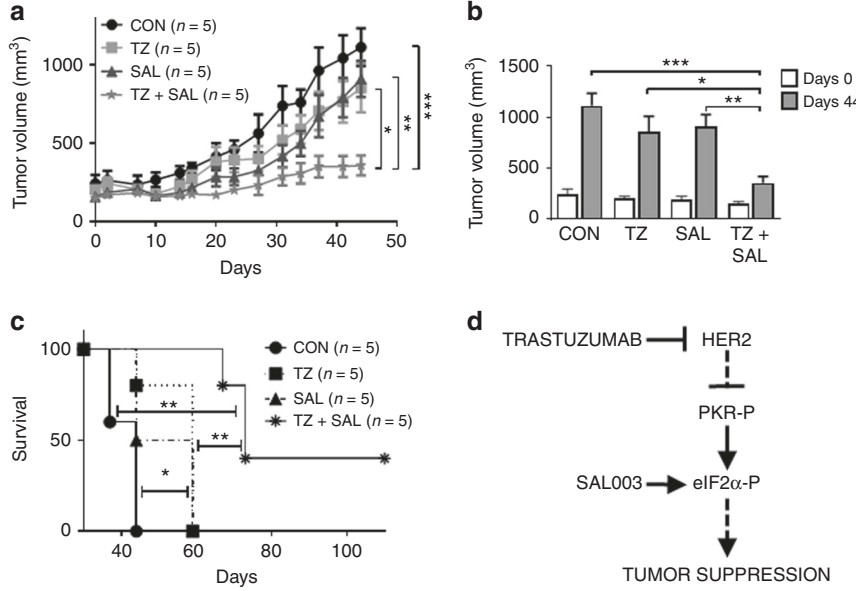

**Fig. 6** SAL003 sensitizes HER2+ gastric cancer PDXs to Trastuzumab therapy in mice. **a** Total of 20 NOG mice were engrafted with a patient derived Trastuzumab-resistant HER2+ gastric cancer into the flanks subcutaneously and randomized into 4 treatment groups ($n = 5$ per group). Mice were subjected to intraperitoneal injections of 0.9% NaCl (CON), 5 mg/kg/week Trastuzumab (TZ), intratumor injections of 1.5 mg/kg/day SAL003, as well as combined TZ and SAL003 at the same concentrations as the single treatments. Tumor growth was monitored for the indicated times. **b** Tumor volume (mm$^3$) in NOG mice assessed at day 0 and 44 of drug treatments. **c** Survival of NOG mice bearing HER2+ gastric tumors throughout the course of the experiment. **a–c** Data represent mean ± SEM *$p < 0.05$; **$p < 0.01$; ***$p < 0.001$. **d** The PKR/eIF2α-P arm is induced by Trastuzumab in HER2+ tumors that are sensitive to treatment. In Trastuzumab-resistant tumors, stimulation of eIF2α-P by treatment with the eIF2α-phosphatase inhibitor SAL003 exhibits anti-tumor effects and sensitizes resistant tumors to Trastuzumab therapy

IHC analyses of the TMAs indicated the presence of high levels of mainly cytoplasmic eIF2α-P in 50% of tumor specimens (Fig. 7b). When we assessed the relationship between eIF2α-P and clinicopathological characteristics of HER2+ patients, we found no significant correlation between eIF2α-P and age, tumor size, tumor grade, lymph node status, or lympho-vascular invasion, as well as no correlation between eIF2α-P and either estrogen receptor (ER) or progesterone receptor (PR) status. Low eIF2α-P was independent of HER2 expression whereas high eIF2α-P correlated with increased HER2 levels (H-score 3+) ($p = 0.0111$, Table 1).

In the same cohort of metastatic cases, we examined the relationship between eIF2α-P and patient response to Trastuzumab-based therapy after at least 9 weeks of treatment. The objective response rate (ORR) included complete response (CR), partial response (PR) and stable disease (SD) which were grouped together and compared with progressive disease (PD). Notably, patients with low eIF2α-P tumors showed lower ORR and higher PD rate after Trastuzumab treatment than patients with high eIF2α-P (Table 2). Furthermore, a significantly higher proportion of patients with tumors expressing low eIF2α-P [18 of 24 (75%)] progressed under Trastuzumab in less than 6 months, as compared with patients with high eIF2α-P [6 of 24 (25%); Table 2]. Consistently, a significantly greater percentage of patients with low eIF2α-P experienced less than 1 and 2 year-survival following Trastuzumab treatment [16 of 24 (67%), and 22 of 24 (92%), respectively], as compared with patients with high eIF2α-P [8 of 24 (33%), and 15 of 24 (62%), respectively; Table 2]. Moreover, the time to tumor progression (TTP) and overall survival (OS) of patients with low eIF2α-P tumors (median 3.6 and 9.3 months, respectively) was significantly lower than those of patients with high eIF2α-P tumors after Trastuzumab-based therapy (median 7.6 and 17.8 months, respectively) (Table 2). These data suggested that low eIF2α-P correlates with poor

clinical response to Trastuzumab therapy in the HER2+ breast cancer patients.

We also assessed the prognostic value of eIF2α-P and its correlation with TTP and OS. Kaplan–Meier survival analysis showed that high eIF2α-P is significantly associated with longer TTP ($p = 0.0204$) and improved OS ($p = 0.0429$) in the HER2+ breast cancer patients (Fig. 7c). Moreover, in multivariate analysis, high eIF2α-P emerged as an independent prognostic factor for long TTP (HR, 0.392; CI, 0.166–0.687; $p = 0.0027$). Similarly, high eIF2α-P emerged as an independent positive prognostic factor for OS (HR, 0.374; CI, 0.171–0.819; $p = 0.0140$) (Table 3). These data indicated that high eIF2α-P is an independent positive prognostic factor for the response of the HER2+ breast cancer patients to Trastuzumab treatment.

## Discussion
We demonstrate that the PKR/eIF2α-P arm exhibits a tumor suppressor function in a mouse model of HER2+ breast tumorigenesis. The anti-tumor function of the arm depends on ATF4, which increases the expression and activity of anti-proliferative proteins like P21$^{CIP1}$ and phosphorylated JNK1/2 (Fig. 4e). Former studies demonstrated the anti-tumor properties of P21$^{CIP1}$ in different mouse models of breast cancer[30]. In humans, polymorphisms of CDKN1A, which encodes for P21$^{CIP1}$, have been linked to increased risk of advanced breast cancer[31]. Also, loss of expression or increased cytoplasmic localization of P21$^{CIP1}$ is associated with reduced survival and predicts a significantly poorer outcome in response to adjuvant Trastuzumab therapy[32,33]. Phosphorylation of JNK1/2 downstream of ATF4 is mediated by the suppression of DUSP1, which plays an important role in the anti-tumor effects of PKR in mouse NEU breast tumors (Fig. 4). Several studies have supported a context-dependent function of DUSP1 in cancer. That is, DUSP1 promotes prostate, colon, bladder, gastric, breast and lung cancer,

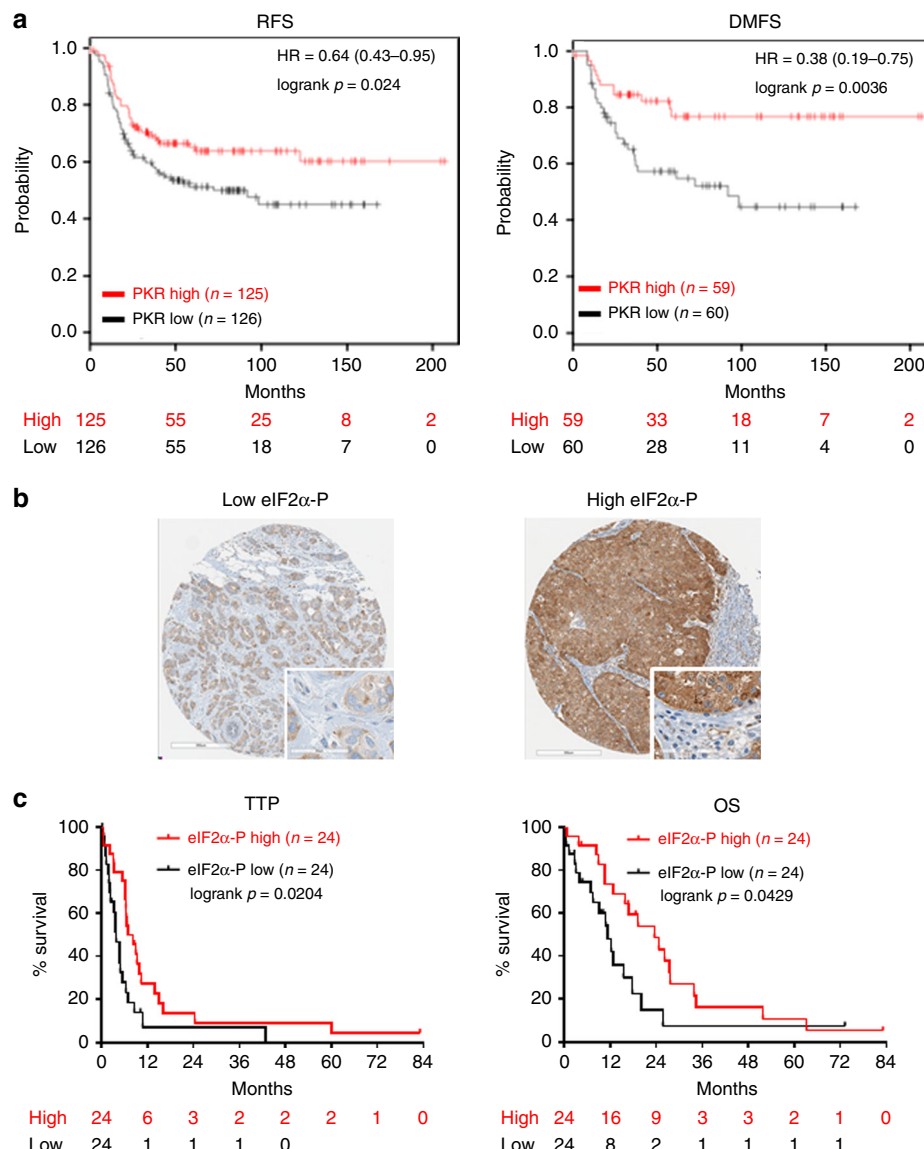

**Fig. 7** PKR mRNA and eIF2α-P prognosticate a better response of HER2+ breast cancers to therapy. **a** Kaplan–Meier curves of PKR mRNA expression for relapse-free survival (RFS) and distant metastasis free survival (DMFS) of HER2+ breast cancer patients subjected to neoadjuvant chemotherapy (www.kmplot.com; Gene ID: 204211_x_at, Intrinsic subtype: HER2+). **b** Representative IHC staining in breast tumor cores of HER2+ patients with low or high eIF2α-P. Stromal cells were negative for eIF2α-P as opposed to tumor cells that were positive for eIF2α-P (inserts). Original magnification ×400. **c** Kaplan-Meier survival curves of time to tumor progression (TTP) and overall survival (OS), starting from initiation of Trastuzumab-based chemotherapy therapy, of HER2+ breast cancer patients with low or high eIF2α-P

but it exhibits anti-tumor effects in hepatocellular, as well as head and neck cancers[34]. Also, DUSP1 sensitizes non-small cell lung cancers and pancreatic cancers to chemotherapy, but it promotes chemoresistance in breast and ovarian cancers[35–38]. The stimulation of JNK1/2 phosphorylation accounts for the anti-proliferative effects of PKR in mouse NEU tumors (Fig. 4). This finding agrees to previous studies demonstrating the anti-tumor properties of JNK1/2 in different mouse models of breast cancer. Specifically, JNK1/2 prevents spontaneous breast tumor development in mice heterozygous for P53 and inhibits tumor development in transgenic mice expressing the polyoma middle T oncogene in the mammary gland[39,40]. Also, JNK1/2 deficiency promotes tumor formation in a KRAS/P53 mouse model of breast cancer[41], whereas inactivation of MKK4 or MKK7, which are upstream activators of JNK1/2, is associated with oncogenic transformation in the mouse mammary gland[42,43].

Tumors with a heterozygous eIF2αS51 knock-in mutation (eIF2α$^{S/A}$) displayed a detectable but statistically insignificant trend to grow better than PKR$^{-/-}$ tumors in NEU mice (Fig. 1c, $p = 0.0394$). This data raised the interesting question about the role of eIF2α kinases other than PKR in the inhibition of HER2+ breast cancer. PERK assumed a pro-tumorigenic role in our mouse model of HER2+ breast cancer through its ability to stimulate the anti-oxidant function of nuclear factor (erythroid-derived)-like-2 (NFE2L2)[44]. The ability of PERK to act via eIF2α-P-independent pathways was also supported by findings showing its pro-metastatic function in triple negative breast tumors through the action of cAMP responsive element binding protein 3-like 1 (CREB3L1)[45]. GCN2 has been implicated in pro-tumorigenic pathways in solid tumors[5], but its function in HER2+ cancers remains to be established. On the other hand, pharmacological activation of HRI has been linked to anti-tumor

**Table 1 Clinicopathologic characteristics of HER2+ breast cancer patients according to eIF2α-P levels**

| Clinico-pathologic characteristic | eIF2α-P low (H-score ≤ 101.165) N = 24 | | eIF2α-P high (H-score > 101.165) N = 24 | | |
| --- | --- | --- | --- | --- | --- |
| | Number of patients | Percentage | Number of patients | Percentage | P value |
| Age at diagnosis (years) | | | | | |
| ≤50 | 10 | 41.67 | 15 | 62.50 | 0.1486 |
| >50 | 14 | 58.33 | 9 | 37.50 | |
| Tumor size (cm) | | | | | |
| T0 (0–2) | 10 | 41.67 | 5 | 20.83 | 0.1195 |
| T1 (>2) | 14 | 58.33 | 19 | 79.17 | |
| Tumor grade | | | | | |
| Low | 0 | 0.00 | 0 | 0.00 | |
| High | 24 | 100.00 | 24 | 100.00 | |
| LN status | | | | | |
| Negative | 7 | 29.17 | 5 | 20.83 | 0.5050 |
| Positive | 17 | 70.83 | 19 | 79.17 | |
| LVI | | | | | |
| Negative | 8 | 33.33 | 6 | 25.00 | 0.5254 |
| Positive | 16 | 66.67 | 18 | 75.00 | |
| ER status | | | | | |
| Negative | 11 | 45.83 | 10 | 41.67 | 0.7711 |
| Positive | 13 | 54.17 | 14 | 58.33 | |
| PR status | | | | | |
| Negative | 14 | 58.33 | 14 | 58.33 | 1.0000 |
| Positive | 10 | 41.67 | 10 | 41.67 | |
| HER2 status | | | | | |
| IHC 2+ | 11 | 45.83 | 3 | 12.50 | 0.0111* |
| IHC 3+ | 13 | 54.17 | 21 | 87.50 | |

LN lymph node, LVI lympho-vascular invasion, ER estrogen receptor, PR progesterone receptor, HER2 human epidermal growth factor receptor 2
*p < 0.05

**Table 2 HER2+ breast cancer patient response to Trastuzumab based on eIF2α-P levels**

| | eIF2α-P low, N = 24 | | eIF2α-P high, N = 24 | | |
| --- | --- | --- | --- | --- | --- |
| | Number of patients | % | Number of patients | % | P value |
| Tumor response | | | | | |
| ORR (SD + CR + PR) | 15 | 62.50 | 21 | 87.50 | 0.0455* |
| PD | 9 | 37.50 | 3 | 12.50 | |
| Progression | | | | | |
| TTP < 6 months | 18 | 75.00 | 6 | 25.00 | 0.0005*** |
| TTP > 6 months | 6 | 25.00 | 18 | 75.00 | |
| Median TTP (month) | 3.6 | | 7.6 | | 0.0006*** |
| Site of progression | | | | | |
| Visceral (lung + liver) | 7 | 43.75 | 12 | 54.54 | 0.5595 |
| Bone | 5 | 18.75 | 5 | 18.18 | |
| Brain | 3 | 18.75 | 6 | 22.73 | |
| Other | 3 | 18.75 | 1 | 4.55 | |
| Survival | | | | | |
| OS < 1 year | 16 | 66.67 | 8 | 33.33 | 0.0254* |
| OS > 1 year | 8 | 33.33 | 16 | 66.67 | |
| OS < 2 years | 22 | 91.67 | 15 | 62.50 | 0.0363* |
| OS > 2 years | 2 | 8.33 | 9 | 37.50 | |
| Median OS (month) | 9.3 | | 17.8 | | 0.0223* |

ORR objective response rate, SD stable disease, CR complete response, PR partial response, PD progressive disease, TTP time to tumor progression, OS overall survival. P values ≤ 0.05 were considered statistically significant
*p < 0.05; **p < 0.01; ***p < 0.001

effects in breast tumors indicating its possible tumor suppressive function in this form of cancer[46,47].

HER2 downregulation increased PKR T446 autophosphorylation and eIF2α-P in HER2+ breast tumor cells suggesting an inhibitory role of HER2 on PKR (Supplementary Fig. 5a). How HER2 inhibits PKR activity is not immediately clear. PKR might be regulated by kinases and/or phosphatases in HER2+ tumors considering its ability to be phosphorylated by receptor tyrosine kinases and receptor-associated tyrosine kinases[48–50]. It is also possible that HER2 signaling impacts on the expression of RNAs that bind to and regulate PKR activity. In this regard, inverted Alu repeats in the 3' untranslated region (3' UTR) of mRNAs have been shown to bind to and activate PKR[51,52]. Trastuzumab upregulates the PKR-eIF2α-P arm and the downstream anti-tumor pathways in sensitive but not resistant breast tumor cells (Supplementary Fig. 5b). Treatment with the

**Table 3 Multivariate Cox regression analysis for TTP and OS of HER2+ breast cancer patients**

| Parameter | Time to tumor progression (TTP) | | | Overall survival (OS) | | |
|---|---|---|---|---|---|---|
| | HR | 95% CI | P value | HR | 95% CI | P value |
| eIF2α-P — high | 0.392 | 0.166–0.687 | 0.0027* | 0.374 | 0.171–0.819 | 0.0140* |
| Tumor size (cm) — T1 (>2) | 2.386 | 1.042–5.462 | 0.0397* | 2.535 | 1.001–6.422 | 0.0499* |
| Lymph node (LN) status — positive | 1.523 | 0.683–3.392 | 0.3035 | 0.711 | 0.316–1.602 | 0.4108 |
| Lympho-vascular invasion (LVI) — positive | 0.540 | 0.258–1.131 | 0.1024 | 0.661 | 0.292–1.495 | 0.3200 |
| Estrogen receptor (ER) status — positive | 1.279 | 0.573–2.855 | 0.5489 | 0.930 | 0.408–2.119 | 0.8631 |
| Progesterone receptor (PR) status — positive | 0.797 | 0.363–1.746 | 0.5702 | 0.720 | 0.312–1.662 | 0.4411 |

*HR* hazard ratio, *CI* confidence interval
*p < 0.05; **p < 0.01; ***p < 0.001

eIF2α-phosphatase inhibitor SAL003 inhibits the colony forming efficiency of the breast tumor cells and substantially potentiates the anti-proliferative and anti-tumor effects of Trastuzumab on HER2+ breast and gastric tumors in culture and immune deficient mice (Fig. 5, 6; Supplementary Fig. 7). These findings support the interpretation that stimulation of eIF2α-P by SAL003 overcomes resistance of HER2+ cancers to Trastuzumab (Fig. 6d), but it is currently unclear whether this process depends on the upregulation of P21[CIP1] and JNK1/2 activity in the treated tumors.

The anti-tumor function of sustained eIF2α-P is supported by a previous study showing that disruption of the acinar morphogenesis of human epithelial MCF10A cells by HER2 in 3D cultures is prevented by treatment by the eIF2-phosphatase inhibitor Salubrinal[53]. Also, Salubrinal alone or in combination with other targeted therapies elicits anti-tumor effects in tumors of different origin including multiple myeloma and hepatocellular carcinoma[54,55]. Because Salubrinal or the more powerful derivative SAL003 does not cause significant toxicity in mice[56,57], the use of eIF2α-phosphatase inhibitors may prove a suitable approach to increase the efficacy of Trastuzumab therapy in the clinics.

We show that increased eIF2α-P emerges as a potential independent prognostic value for tumor progression and overall survival of HER2+ metastatic breast cancer patients treated with Trastuzumab-based chemotherapy. Interestingly, low eIF2α-P levels were independent of HER2 expression whereas high eIF2α-P levels correlated with increased HER2 expression in the metastatic breast tumors (Table 1). BT474 cells that became resistant to Trastuzumab exhibited elevated levels of HER2 and eIF2α-P consistent with the data from the clinical samples (Fig. 5a; Supplementary Fig. 5b). However, treatment of sensitive, as well as resistant BT474 cells with SAL003 did not influence HER2 expression suggesting that increased eIF2α-P does not impact on HER2 (Fig. 5a). Perhaps pathways acting in parallel with eIF2α-P influence HER2 expression in the breast tumors. As with our findings, a recent study indicated that elevated eIF2α-P correlated with better disease-free survival and served as an independent prognostic factor in patients with triple-negative breast cancers[58]. However, the significance of increased eIF2α-P in the development and treatment of triple negative breast tumors remains to be determined. Also, increased eIF2α-P was shown to correlate with a higher influx of tumor infiltrating lymphocytes in HER2+ breast cancer specimen[59], but the clinical relevance of this observation is unknown. Albeit several biomarkers of Trastuzumab therapeutic response have been previously identified in HER2+ breast cancer, HER2 remains the only validated biomarker with demonstrated clinical utility so far, despite its low positive predictive value[60]. Our findings may have potential implications for treatment decisions and clinical follow-up of HER2+ breast cancer patients. The predictive value of eIF2α-P for Trastuzumab therapeutic response in HER2+ breast cancer may

justify clinical trials to determine the therapeutic utility of SAL003 and possibly other eIF2α-phosphatase inhibitors for sustained induction of eIF2α-P in combination with Trastuzumab-based therapies. These may prove powerful approaches for the implementation of personalized targeted therapies against breast and other forms of HER2+ cancers.

## Methods

**Cell culture and treatments**. BT474 cells were obtained from the American Type Culture Collection (ATCC) (Rockville, MD). The BT474 cells were maintained in Dulbecco's modified Eagle's medium (DMEM; Wisent) with 10% fetal bovine serum (FBS, Wisent) and antibiotics (penicillin/streptomycin, 100 units/ml; Life Technologies). The primary BM-132 cells were derived from a HER2+ breast cancer patient pre-treated with chemotherapy and developed resistance to Trastuzumab. The BM-132 cells were grown in 66% DMEM high glucose (Wisent), 25% Ham's F12 nutrient mix (Invitrogen), 7.5% FBS (Wisent), 0.4 ug/ml Hydrocortisone (Stem cell technologies), 5 ug/ml Insulin (Sigma), 8.4 ng/ml Cholera Toxin (Sigma), 10 ng/ml EGF (Invitrogen), 10 μM Y-27632 RHO/ROCK pathway inhibitor (Stem cell technologies), 100 unit/ml Penicillin and 100 unit/ml Streptomycin (Gibco), 1.48 mM L-glutamine (Wisent). The mouse NEU tumor cells were maintained in the same media supplemented with Mammary Epithelial Growth Supplement (Life Technologies) consisting of 0.4% v/v bovine pituitary extract (BPE), 1 μg/ml recombinant human insulin-like growth factor 1, 0.5 μg/ml hydrocortisone and 3 ng/ml human epidermal growth factor. Cells were tested for mycoplasma contamination. SAL003 was purchased from Sigma-Aldrich (# S4451) whereas Trastuzumab was manufactured by Roche and provided by the Pharmacy of Jewish General Hospital (JGH).

**Patient-derived tumors and tumorigenic assays in mice**. A patient-derived xenograft was generated from a surgical resection of a gastric adenocarcinoma poorly differentiated (Histological grade 3). The patient had Stage IV gastric cancer with liver and bone metastasis and was treated with Herceptin alone. The residual tumor post treatment expressed high levels of ERBB2, which was confirmed by HerceptTest (Score 3+). The HER2+ gastric cancer PDX was generated by the subcutaneous implantation of small pieces of fresh tissue into the flanks of NOG (NOD.Cg-Prkdcscid Il2rgtm1Sug/JicTac) mice (Taconic). Anti-tumor treatments of tumor-bearing NOG mice were initiated when tumor volume reached 100 mm$^3$. For testing the tumorigenicity of the mouse NEU tumor cells, cells were suspended in 1:1 v/v mixture of phosphate buffer saline (PBS): Matrigel (Corning) and injected subcutaneously into the flanks of SCID mice (Charles River Inc.). Mice received 2 injections of $5 \times 10^5$ cells in 100 μl volume of PBS:Matrigel mix. Tumor growth in NOG and SCID mice was measured with digital calipers two times per week, and the volume calculated by the formula: tumor volume [mm$^3$] = π/6 × (length [mm]) × (width [mm]) × (height [mm]). The animal studies were performed in accordance with the Institutional Animal Care and Use Committee (IACUC) of McGill University and procedures were approved by the Animal Welfare Committee of McGill University (protocol #5754).

**DNA transfection and lentivirus infection**. DNA transfections were performed with Lipofectamine 2000 reagent (Invitrogen) according to the manufacturer's specifications. CRISPR/CAS9-mediated depletion of mouse PKR was performed by the co-expression of gRNA in Supplementary Table 1 and CAS9 from pST1374-NLS-flag-linker-CAS9 vector[61]. Stable pools of cells expressing the ATF4 or DUSP1 shRNAs listed in Supplementary Table 1 were generated by infection with pLKO.1 lentiviruses and selection at 2.5 μg/ml puromycin. Colony formation assays were performed with 10$^4$ cells subjected to anti-tumor treatments as indicated in the corresponding figures for 14 days. Cells were fixed in 3.7% v/v formaldehyde and stained with 0.2% w/v crystal violet. Colonies were scored using an automated cell colony counter (GelCount; Oxford Optronix).

**RNA isolation and real time PCR.** Total RNA (1 μg) isolated by TRIzol (Invitrogen) was subjected to reverse transcription (RT) with 100 μM oligo (dT) primer using the SuperScript III Reverse Transcriptase kit (Invitrogen) according to the manufacturer's instruction. Real time (quantitative) PCR was performed using the SensiFast SYBR Lo-ROX kit (Bioline) with primers listed in Supplementary Table 1. The qPCR assays included primers for mouse GAPDH and actin mRNAs as internal controls according to the Minimum Information for Publication of Quantitative Real-Time PCR Experiments (MIQE) guidelines[62].

**Protein extraction and immunoblotting.** Cells were washed twice with ice-cold phosphate-buffered saline and proteins were extracted in ice-cold lysis buffer containing 10 mM Tris-HCl, pH 7.5, 50 mM KCl, 2 mM $MgCl_2$, 1% Triton X-100, 3 μg/ml aprotinin, 1 μg/ml pepstatin, 1 μg/ml leupeptin, 1 mM dithiothreitol, 0.1 mM $Na_3VO_4$, and 1 mM phenylmethylsulfonyl fluoride. Extracts were kept on ice for 15 min, centrifuged at $10,000 \times g$ for 15 min (4 °C), and supernatants were stored at −80 °C. Proteins were quantified by Bradford assay (Bio-Rad). Expression of different proteins was tested in parallel by loading 50 μg of protein extracts from the same set of samples on two identical sodium dodecyl sulfate (SDS)-polyacrylamide gels. After protein transfer to Immobilon-P membrane (Millipore), the two identical blots were cut into smaller pieces based on the size of proteins to be tested. One piece was probed for the phosphorylated protein of interest whereas the other identical piece for the corresponding total protein. The antibodies used for immunoblotting are listed in Supplementary Table 2. Proteins were visualized by enhanced chemiluminescence (ECL) according to the manufacturer's specification (Amersham Biosciences).

**Preparation of tissue microarrays (TMAs).** Specimens from HER2+ metastatic breast cancer (MBC) patients treated with Trastuzumab-based chemotherapy were obtained from the Alberta Cancer Registry in Canada between 1998 and 2002[29,63]. In this study, we reviewed the original HER2 immunohistochemistry studies and performed chromogenic in situ hybridization on all cases, as per the published guidelines for HER2 testing[64]. Patients with 2+ immunohistochemistry scores and no HER2 amplification were excluded. Formalin-fixed paraffin-embedded tumor tissue specimens were retrieved from the patients. Tissue microarrays (TMAs) were constructed using either duplicate, triplicate, or quadruplicate cores from each patient's tumor. TMAs also included control cores from normal tonsil tissue. TMA sections of paraffin-embedded tissue were processed for immunohistochemistry. Tissues were cut at 4-μm, placed on SuperFrost/Plus slides (Fisher) and dried overnight at 37 °C, before IHC processing. The slides were then loaded onto the Discovery XT Autostainer (Ventana Medical System).

**Histology and immunohistochemistry.** Mouse tissues were fixed in 10% buffered formalin phosphate, paraffin embedded and sectioned. Paraffin was removed from the sections after treatment with xylene, rehydrated in graded alcohol, and used for H&E staining and immunostaining. Antigen retrieval was performed in sodium citrate buffer. Primary antibodies were incubated at 4 °C overnight (antibodies listed in Supplementary Table 2). Secondary antibodies listed in Supplementary Table 2 were incubated at room temperature for 90 min. Sections were counterstained with 20% Harris modified hematoxylin (Thermo Fisher Scientific), mounted in Permount solution (Thermo Fisher Scientific) and scanned using an Aperio Scanscope AT Turbo scanner (Leica biosystems). Quantification of stained sections was performed using Aperio Imagescope software (Leica Biosystems).

Immunohistochemistry of eIF2α-P in human specimens was performed on a Discovery XT Autostainer (Ventana Medical System). All solutions used for automated immunohistochemistry were from Ventana Medical System. Slides underwent de-paraffinization, heat-induced epitope retrieval (CC1 prediluted solution Ref: 950-124, standard protocol). Immunostaining of eIF2α-P was performed with the mouse monoclonal anti-eIF2α-P (Abcam, Cat # ab32157), which was diluted at 1:100 in the antibody diluent (Ventana) and manually applied for 32 min at 37 °C followed by the appropriate detection kit (Omnimap anti-mouse HRP Ref: 760–4310 and ChromoMap-DAB Ref: 760–159). The omission of the primary antibody was used as negative control. Slides were counterstained with hematoxylin for four minutes, treated with Bluing Reagent for 4 minutes, removed from the autostainer, washed in warm soapy water, dehydrated through graded alcohols, cleared in xylene, and mounted with Permount. Quantification of eIF2α-P-stained sections was performed using Aperio Imagescope software (Leica biosystems). The same pattern of staining but of weaker intensity was obtained with an anti-eIF2α-P antibody from Cell Signaling Tech. (Cat# 3597) (Supplementary Fig. 8). Staining of eIF2α-P was quantified using a four-value intensity score (0 for negative, 1 for weak, 2 for moderate, or 3 for strong), and the percentage of positive tumor cells. Final scoring was evaluated using the histoscore (H) obtained by the formula: (3× percentage of "3+" strongly staining tumor cells) + (2× percentage of "2+" moderately staining tumor cells) + (percentage of "1+" weakly staining tumor cells); giving a range of 0 to 300. The average H-score obtained among different cores of the same tumor was used as the final score of that tumor in the statistical analysis.

**Ethical regulations.** The Trastuzumab-resistant HER2+ gastric cancer patient consented to sample collection for the central biobank of the JGH (protocol

#10-153) and PDX generation was performed under protocol (#14-168). The use of HER2+ breast tumor samples from patients subjected to Trastuzumab treatments was approved with protocol #17-150. The patient-derived HER2+ breast cancer BM-132 cells were part of the breast cancer biobank of the JGH (protocol #05-006). All protocols were approved by the JGH Research Ethics Committee. Specimens from HER2+ metastatic breast cancer patients were obtained following the ethical guidelines implemented by the Alberta Cancer Board Ethics Review Board.

**Statistical analysis.** The median H-score of all tumors was selected as the cut-off point for eIF2α-P. Tumors were dichotomized into low-level and high-level groups as follows: eIF2α-P: low (H-score ≤ Median 101.165), high (H-score > Median 101.165). The variables analyzed included eIF2α-P, age at diagnosis, tumor size, lymph node, lympho-vascular invasion, ER, PR, and HER2 expression. The end points of this study were time to tumor progression (TTP) defined as the time from initiation of Trastuzumab until the date of tumor progression, and overall survival (OS) defined as the time from initiation of Trastuzumab until the date of death, or date of last follow-up for live patients.

Patients were evaluated for a response after at least 9 weeks of Trastuzumab and then at 12-weeks intervals using the Response Evaluation Criteria in Solid Tumors assessment[65]. Complete response (CR) was defined as the disappearance of all target lesions. Partial response (PR) was defined as a decrease of more than 50% in the dimensions of all measurable lesions. Progressive disease (PD) was defined as an increase of more than 25% in the dimensions of any measurable lesion. Stable disease (SD) was defined by neither PR nor PD criteria met, typically involving a small amount of growth or a small amount of shrinkage (<20%). The objective response rate (ORR) was defined as CR + PR + SD.

Chi-square test and Fisher's exact test were used, when appropriate, to evaluate the association of eIF2α-P expression with clinico-pathologic characteristics, tumor response, progression, and survival. Wald test was used to evaluate differences between medians of TTP and OS. TTP and OS curves were constructed according to the Kaplan–Meier method. The log-rank test was used to compare patient survival probability between eIF2α-P groups. The multivariate Cox proportional hazards regression model was used to calculate the effect of eIF2α-P expression on TTP and OS, with adjustments made for known prognostic factors (tumor size, lymph node, lympho-vascular invasion, ER and PR expression). All analyses were two-sided with $p \leq 0.05$ being considered significant. SAS 9.4 (SAS Institute Inc., Cary, NC, USA) was used to conduct the calculations. Error bars represent standard error as indicated and the significance in differences between arrays of data was determined using two-tailed Student $T$ test.

**Reporting summary.** Further information on research design is available in the Nature Research Reporting Summary linked to this article.

## Data availability

All data in this study are available within the Article and Supplementary Information or from the corresponding author on reasonable request. The Kaplan–Meier (KM) plots of the gene expression data were obtained from www.kmplot.com, KM plotter for breast cancer, gene ID: PKR, EIF2AK2, 204211_x_at, selection criteria of intrinsic subtype:HER2+.

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

## Acknowledgements

We thank C. Patsis for assistance in mouse breeding; S. Kane (Beckman Research Institute of City of Hope) for Trastuzumab-resistant BT474 cells; J. C. Bell (Ottawa Regional Cancer Centre) for PKR⁻/⁻ mice and R. Kaufman (Sanford Burnham Medical Research Institute) for eIF2α$^{S/A}$ mice. The work was supported by grants from Quebec Breast Cancer Foundation and Canadian Institutes of Health Research (CIHR MOP-

153388) to A.E.K., grants from Fonts de Recherche du Québec-Santé (FRQS) Reseau de Cancer, Axe cancer du sein/ovaire and McPeak Sirois Consortium and Jewish General Hospital Foundation to M. Basik, a grant from Montreal General Hospital Foundation to S.S. and a grant from CIHR (MOP-111143) to J.U.-S.

## Author contributions

A.E.K. designed the study, analyzed the data, wrote and edited the manuscript. C.D. collected and analyzed data in Figs. 1, 2, 3, 4, 5a, b, 6a–c, Supplementary Fig. 1–8, partially wrote and edited the manuscript. N.G. collected data in Figs. 1d, 2d, 3a, 4a, 5a, Supplementary Figs. 1, 4, 5, 7. J.K. collected data in Supplementary Figs. 1, 4, 5, 7. S.W. contributed to data collection in Fig. 1a–c, Fig. 2a–c and Supplementary Fig. 2. C.C. collected and interpreted data in Fig. 5b,c. M. Buchanan contributed to data collection in Fig. 6. A.A.-M. contributed to data collection in Fig. 6a and Supplementary Fig. 6. S.S. contributed to study design for the prognostic value of eIF2α-P and together with G.A., B.A., and J.D. performed the analyses of data in Fig. 7b, c and Tables 1–3. J.T. contributed to data analysis in Fig. 7b and Supplementary Fig. 6. J.U.-S. contributed to study design, data analysis in Figs. 1a–c, 2a, Supplementary Fig. 2, provided reagents and expertize. M. Basik contributed to data analysis of Fig. 6a–c, Supplementary Fig. 6, provided reagents and expertize.

## Additional information

**Competing interests:** All authors declare no competing interests.

