## [Peer Review File · Nature Communications]

Reviewers' comments:

Reviewer #1 (Remarks to the Author):

This work assessed a potential role of PKR-PeIF2 α -ATF4 pathway in breast tumorigenesis and chemoresistance. The authors used genetic approaches to demonstrate that PKR, as well as PeIF2 α , inhibit tumorigenesis in a mouse model of HER2+ breast cancer. The underlying mechanism involves ATF4 expression required for downstream upregulation of p21 and activation of JNK apoptotic signaling pathway. They also reported that PKR and PeIF2 α are upregulated by Trastuzumab in resistant HER2+ breast tumor cells and HER2+ breast tumors including PDX. While this study has the merit to highlight the possibility of targeting PeIF2 α to prevent resistance of breast cancer to conventional treatment, it suffers from being based mostly on correlative results lacking mechanistic data. Patient data constitute the power of this study but unfortunately, it relies on the use of a single antibody. The authors need also to clearly discuss their results relative to those (e.g., PMID: 29057869; 27272779) describing a possible pro-breast cancer role of PeIF2 α and upstream stress kinase.

Specific concerns:

1-Fig 1a-b: As the authors are well aware, the level of PKR mRNA do not necessarily reflect the level of PKR or its activity! Would it be possible to document the level of tumoral PKR in order to conclude its possible implication on tumors growth and metastasis?

2- Fig 1e: These data show that tumors from eIF2 α SA are bigger than PKR helicase defective tumors. Please discuss these data, e.g., is it possible that other eIF2 α kinase might be involved?

3-Fig. 1f: The amounts of PeIF2 α and downstream ATF4 in tumors derived from mice expressing wt PKR and eIF2 α are strikingly high! Is this is specific to the mice strain used in this study or this is reflecting a constitutive activation of the PKR-PeIF2 α pathway in tumors? In any case, it would be helpful if the authors could assess the level of PKR and PeIF2 α in non-tumoral breast samples isolated from the same animals. Also, the high level of PeIF2 α shown in wt tumors samples is likely to affect general translation initiation thus reprogramming the translatoome. This point should be discussed.

4-Suppl. Fig. 2a. They showed that the expression of a HER2 activated form results in an inhibition of the activation of PKR and downstream PeIF2 α in breast epithelial MCF10A cells. How mechanistically HER2 could inhibit PKR activity? They also showed that HER2 inhibition by Trastuzumab increased PKR T446 phosphorylation and PeIF2 α in BT474 breast cancer cells, which were sensitive (BT474S) to Trastuzumab. There is no evidence, however, that Trastuzumab activates PKR-PeIF2 α by inhibiting HER2.

It is also not clear to this reviewer why Trastuzumab affects PKR-PeIF2 α in BT474S and not in BT474R. Because BT474R do not show an activation of the PKR-PeIF2 α -ATF4-p21-JNK pathway(s) in response to Trastuzumab, the authors should verify if this pathway is defective in this cell line by subjecting these cells to specific stresses that are known to activate PKR and induce ATF4. If the resistance of BT474 to Trastuzumab is due to the constitutive activation of that pathway, then its downregulation should sensitize them to treatment! Please verify if this is the case.

Trastuzumab seems to significantly affect AKT phosphorylation in BT474S. This effect is not evident however in Fig. 6a. Please provide quantification of those W. blot data. The results presented in Figs 6 and S2 would also suggest that the sustained activation of AKT in BT474 promotes resistance despite constitutive activation of PKR pathway. In this case, enforcing the expression of activated AKT in sensitive BT474 should render them resistance to Trastuzumab.

5-Fig 3: The authors identified ATF4 as the downstream target and effector of PKR-PeIF2 α

pathway. What is the effect of depleting ATF4 on the proliferation/death of their breast cancer cells and tumors models? The same question applied to p21. There is a clear effect of ATF4 depletion on p21 expression. No evidence is however provided demonstrating that this ATF4-mediated effect occurs at the transcriptional level.

6-They showed that breast tumors derived from Trastuzumab-treated patients have more PeIF2 α than those from the same patients before treatment. There is no mention however if those patients resist or not treatment! They could not verify PKR activation because of the lack of suitable antibodies. Why they did not check the level of the downstream effectors ATF4 and p21.

7-They addressed the prognostic value of PeIF2 α from the analyses of TMAs derived from patients with HER2+ metastatic breast cancer. Low PeIF2 α was independent of HER2 expression whereas high PeIF2 α seems to correlate with increased HER2 levels. Since PeIF2 α is known to modulate the proteome, it will be helpful to address if the high level of PeIF2 α could affect HER2 translation. Although not mentioned, I expect that the data that are generated from these analyses rely on the use of a single anti-PeIF2 α antibody. The use of a second anti-PeIF2 α antibody or antibodies specific to downstream targets is recommended to confirm these important patient studies.

Reviewer #2 (Remarks to the Author):

Darini et. al

An Integrated stress response via PKR suppresses HE2-mediated breast tumorigenesis and increases the efficacy of Trastuzumab therapy

This manuscript demonstrates that PKR/eIF2 α suppresses NEU-mediated breast tumor growth in mice as PKR $^{-/-}$ or eIF2 α s/a mice in MMTV-NEU background had larger tumors than NEU control mice. These tumors have reduced ATF4 and p21 levels, reduced JNK activity but elevated DUSP1 levels. Authors show that PKR/eIF2 α requires ATF4 to regulate p21, DUSP1 and increase JNK activity in NEU breast tumors. Authors also show that increase of DUSP1 level is important in PKR $^{-/-}$ -NEU tumors as reducing DUSP1 levels reduces tumor growth in these mice. Authors also show that Trastuzumab (TZ) treatment increases eIF2 α phosphorylation in HER2+ breast tumor biopsies. Clinical data is presented that suggests treating HER2+ breast tumors increases phospho-eIF2 α and high levels of eIF2 α phosphorylation is associated with increased clinical response to Trastuzumab treatment in HER2+ breast cancer patients. Lastly, authors show that activating eIF2 α with drug SAL003 blocks tumor growth of BT474-TZ-resistant cell lines and enhances TZ effects. They verify these effects in vivo using a HER2+ gastric cancer PDX TZ-resistant model.

Thus, authors conclude that PKR/eIF2 α play a role in HER2+ tumor suppression and activating this arm may potentiate the anti-tumor effects of Trastuzumab.

Overall, the manuscript provides novel data regarding role of PKR/eIF2 α in regulating HER2+ breast cancers and response to Trastuzumab. Manuscript provided novel mechanistic and clinical data regarding phospho-eIF2 α in HER2+ breast cancers. However, there are some issues with use of a gastric cancer model in a breast cancer focused manuscript as well as organization of some of the data that needs to be addressed before publication.

Major issues:

1. Figure 7 uses an in vivo HER2+ gastric cancer PDX model to essentially validate in vitro results using breast cancer cells in Fig. 6. Switching to a gastric model for the last figure is confusing and not congruent with rest of manuscript (or title of manuscript). Authors should perform in vivo experiments with BT474-R cells with these drugs or at minimum show that TZ & SAL regulated pathway (as shown in Fig. 6A) also exists in gastric cancer PDX model (and remove "breast" from title of manuscript).
2. Having the clinical data in middle of manuscript is confusing and made manuscript difficult to

read. It may be better to have it at end of manuscript instead of going back and forth from animal/signaling data (same for abstract).

3. Authors should discuss the discrepancy from clinical data and experimental models. Cell models show that HER2 overexpression reduces phospho- eIF2a and treatment with TZ increases phospho-eIF2a. However, authors found high phospho- eIF2a correlates with high HER2 levels.

Minor issues:

1. Figure 2C: Do these tumors also have reduced KI-67 and increased apoptosis?

2. Figure 3 should be part of Figure 2 (Its in the same section of Results).

3. Fig. 4D. Should show effects of JNK inhibitor on p-JNK in cells (not clear how authors know drug is working).

4. Stats missing from Fig. 7A.

5. Model of Fig. 7D is confusing. Looks like HER2 is activating PKR-P (when its really inhibiting it). Better model is needed.

Reviewer #1

Patient data constitute the power of this study but unfortunately, it relies on the use of a single antibody.

The antibody for phosphorylated eIF2 α in our study is from Abcam (Cat# ab32157). We have compared the Abcam antibody to an antibody from Cell Signaling Technology (CST), which is suitable for immunohistochemistry as per manufacturer's recommendation. Staining of identical human breast tumor sections with the two antibodies resulted in the same pattern. However, the Abcam antibody was of superior quality (please see Suppl. Fig. 8). Previous work from our lab demonstrated the high specificity of the Abcam antibody for phosphorylated eIF2 α in human tumor cells grown in nude mice (Aging 2013;5:884-901). The antibody was previously distributed by Novus and is now provided by Abcam.

The authors need also to clearly discuss their results relative to those (e.g., PMID: 29057869; 27272779) describing a possible pro-breast cancer role of PeIF2 α and upstream stress kinase.

Both published studies concern PERK and not PKR. The study by Kim et al. Anticancer Research (PMID 27272779) shows that increased PERK protein (not activity) and eIF2 α -P are associated with an increased influx of tumor infiltrating lymphocytes in HER2+ breast cancer tissues. The PERK antibody used in the study detects protein but not PERK activity, and therefore, it is uncertain whether eIF2 α -P in these tumors depends on PERK activity. Although the study reports an interesting observation, it does not provide evidence for the clinical significance of it (prognostic and predictive values based on patients' history).

The study by Feng et al. Nature Communications (PMID 29057869) shows a pro-tumorigenic role of PERK in triple negative breast (TNB) tumors through the induction of transcription factor CREB3L1. In this nice study, there is no evidence whether this function of PERK depends on eIF2 α -P. Please note that PERK can function independent of eIF2 α -P as has been demonstrated by its ability to exert pro-tumorigenic effects in breast cancers through the transcription factor NRF2 (Oncogene. 2010;29:3881-95; Ref. 48). In TNB cancers, a different study showed that elevated eIF2 α -P correlated with better disease-free survival and served as an independent prognostic factor (ref. 62). Taken together, the data suggested that the pro-tumorigenic function of PERK in TNB cancer is unlikely to depend on eIF2 α -P.

Data from both papers are now discussed in the revised manuscript (see Discussion and Refs. 49, 63).

Fig 1a-b: As the authors are well-aware, the level of PKR mRNA do not necessarily reflect the level of PKR or its activity! Would it be possible to document the level of tumoral PKR in order to conclude its possible implication on tumors growth and metastasis?

REPLY: Data in Fig. 1a,b, which are now in Fig. 5a,b, were obtained from the analyses of data bases of gene expression profiles of human HER2+ breast tumors from patients subjected to chemotherapy. It remains possible that PKR mRNA expression in the tumor samples was affected by the treatment of patients with chemotherapeutic drugs. The data supported an anti-tumor function of PKR, which was confirmed in the mouse model of HER2+ breast cancer.

We agree with the Reviewer that mRNA levels do not always represent protein expression. Concerning this matter, early studies examined PKR expression and activity in breast tumor cell lines but led to conflicting results [Int J Biochem Cell Biol. 1999;31:175-89; Oncogene. 2000;19:3086-94]. This was largely due to *in vitro* kinase assays of PKR activity, which measured the amount of PKR capable of being activated in the test tubes rather than the amount of PKR that was already activated in the protein extracts. Another reason of the discrepancy was the regulation of PKR by cellular inhibitors in the breast cancer cells, and therefore, its protein levels were not representative of its activity [Oncogene. 2000;19:3086-94]. Based on these findings, we reason that testing PKR protein levels in human tumor samples will not add more to the expression profiles of PKR mRNA.

The development of phosphospecific antibodies made the studies on PKR and other eIF2 α kinases easier and reliable. The best approach to test PKR activity is to detect its autophosphorylation in the tumor samples. However, the T446 antibody used in our studies is of high quality for the immunoblotting of activated PKR in human protein extracts but is not suitable for immunohistochemistry as explained in the original manuscript. Therefore, the only option to test PKR activity in the tumor samples is to measure phosphorylated eIF2 α . Although eIF2 α can be phosphorylated by different kinases, our genetic approach places PKR and phosphorylated eIF2 α in the same anti-tumor pathway, and therefore, we reason that phosphorylated eIF2 α reflects PKR activity in response to Trastuzumab treatments.

Fig 1e: These data show that tumors from eIF2 α SA are bigger than PKR defective tumors. Please discuss these data, e.g., is it possible that other eIF2 α kinase might be involved?

REPLY: The tendency for better tumor growth in eIF2 α ^{S/A} than PKR^{-/-} mice in Fig. 1e (now Fig. 1c, see below) is border line (P=0.0394). Theoretically, other eIF2 α kinases could play a role in the inhibition of breast tumor growth, but this remains to be demonstrated by genetic approaches in mouse models. To date, studies have supported a pro-tumorigenic role of PERK in HER2+ breast tumorigenesis from the analysis of a mouse model similar to one used in our study. However, this function of PERK is independent of phosphorylated eIF2 α and proceeds via the activation of the transcription factor NRF2 (Oncogene. 2010;29:3881-95; Ref. 48). We will include this information in the Discussion of the revised manuscript.

Fig. 1f: The amounts of eIF2 α P and downstream ATF4 in tumors derived from mice expressing wt PKR and eIF2 α are strikingly high! Is this specific to the mice strain used in this study or this is reflecting a constitutive activation of the PKR-PeIF2 α pathway in tumors? In any case, it would be helpful if the authors could assess the level of PKR and PeIF2 α in non-tumoral breast samples isolated from the same animals. Also, the high level of PeIF2 α shown in wt tumors samples is likely to affect general translation initiation thus reprogramming the translatoome. This point should be discussed.

REPLY: We have collected mammary tissue from FVB mice (Normal, WT) and FVB mice expressing the NEU transgene prior to tumor development (Normal, NEU). Also, we collected tumorigenic mammary tissue from FVB mice expressing NEU (Tumor, NEU). We tested for phosphorylated eIF2 α and ATF4 expression in 4 mice from each group.

The data showed that NEU expression decreased the phosphorylation of eIF2 α and ATF4 expression in the tissue prior to tumor formation (Normal, NEU) and exhibited the opposite effect in the tumorigenic tissue (Tumor, NEU) compared to mammary tissue from control mice (Normal, WT). Most likely, this is because tumor forming cells encounter different forms of stress in their microenvironment, which can increase the phosphorylation of eIF2 α and ATF4 expression. This result can explain the increased background levels of

phosphorylated eIF2 α and ATF4 in the mouse breast samples shown in the original manuscript. This data is included in the Results and Suppl. Fig. 1 (plus quantifications) of the revised manuscript.

Suppl. Fig. 2a. They showed that the expression of a HER2 activated form results in an inhibition of the activation of PKR and downstream PeIF2 α in breast epithelial MCF10A cells. How mechanistically HER2 could inhibit PKR activity? They also showed that HER2 inhibition by Trastuzumab increased PKR T446 phosphorylation and PeIF2 α in BT474 breast cancer cells, which were sensitive (BT474S) to Trastuzumab. There is no evidence, however, that Trastuzumab activates PKR-PeIF2 α by inhibiting HER2.

REPLY: We do not presently know how PKR activity is regulated by HER2 in the breast tumor cells. We would like to thoroughly address it in future experiments. Perhaps PKR activity is regulated by phosphorylation in pathways under the control of HER2. It is also possible that PKR binds to specific RNAs in the tumor cells and this function is regulated by HER2. These possibilities are now included in the Discussion of the revised manuscript.

We have tested the effects of HER2 downregulation by siRNAs in BT474 cells. The data show that decreased HER2 results in the activation of PKR and phosphorylation of eIF2 α . This further supports the interpretation that HER2 inhibition contributes to the activation of PKR by Trastuzumab. This data is now included in Suppl. Fig. 5a (plus quantifications).

It is also not clear to this reviewer why Trastuzumab affects PKR-PeIF2 α in BT474S and not in BT474R. Because BT474R do not show an activation of the PKR-PeIF2 α -ATF4-p21-JNK pathway(s) in response to Trastuzumab, the authors should verify if this pathway is defective in this cell line by subjecting these cells to specific stresses

that are known to activate PKR and induce ATF4. If the resistance of BT474 to Trastuzumab is due to the constitutive activation of that pathway, then its downregulation should sensitize them to treatment! Please verify if this is the case.

REPLY: We treated Trastuzumab-sensitive (S) and resistant (R) BT474 cells with doxorubicin, which activates PKR based on previous work from our lab and others (Cell Death Differ. 2011;18:145-54). We found that the PKR-eIF2 α -ATF4-p21-JNK arm was induced in both cell types by doxorubicin. Therefore, the pathway is not defective in the resistant tumor cells. We include this data in the rebuttal letter to demonstrate the activation of the pathway in the resistant tumor cells. However, we decided to exclude them from the revised manuscript to avoid confusion from treatments with two different anti-HER2+ breast cancer treatments (Trastuzumab vs. Doxorubicin).

Data with SAL003 also show the hyperactivation of the PKR-eIF2 α -ATF4-p21-JNK arm in Trastuzumab resistant BT474 cells and HER2+ gastric cancer PDXs in mice (*Figs. 5a and Suppl. Fig. 7*). This is another indication that the anti-tumor pathways downstream of eIF2 α -P are not compromised in Trastuzumab resistant tumors. The data with SAL003 show that constitutive activation of the pathway further suppresses HER2+ tumor growth in combined treatment with Trastuzumab (*Figs. 5,6*).

Trastuzumab seems to significantly affect AKT phosphorylation in BT474S. This effect is not evident however in Fig. 6a. Please provide quantification of those W. blot data.

REPLY:

Quantifications of the data in all Figures are now included.

The results presented in Figs 6 and S2 would also suggest that the sustained activation of AKT in BT474 promotes resistance despite constitutive activation of PKR pathway. In this case, enforcing the expression of activated AKT in sensitive BT474 should render them resistance to Trastuzumab.

REPLY: We tested the effects of activated AKT by SC-79, which is compound that induces conformational changes resulting in the activation of the kinase (PNAS 2012;109:10581-6). We verified the effects of SC-79 on AKT T436 phosphorylation by immunoblotting and tested the colony forming efficacy of BT474 cells treated with SC-79 in the absence or presence of Trastuzumab. We found that activation of AKT did not bypass the anti-survival effects of Trastuzumab. It is well-established that hyper-activation of the PI3K pathway by mutations in PTEN or PI3K contribute to Trastuzumab resistance. Our data support the interpretation that activated AKT is not be enough to counterbalance the susceptibility of the tumor cells to Trastuzumab.

8-Fig 3: The authors identified ATF4 as the downstream target and effector of PKR-PeIF2a pathway. What is the effect of depleting ATF4 on the proliferation/death of their breast cancer cells and tumors models? The same question applied to p21. There is a clear effect of ATF4 depletion on p21 expression. No evidence is however provided demonstrating that this ATF4-mediated effect occurs at the transcriptional level.

REPLY: Downregulation of ATF4 increases the proliferation of mouse NEU breast tumor cells. The data is now included in revised Fig. 3d.

The transcriptional upregulation of P21 by ATF4 was demonstrated in FEBS Lett. 2017;591:3682-91. We have included this information in the revised manuscript (ref. 28).

They showed that breast tumors derived from Trastuzumab-treated patients have more PeIF2a than those from the same patients before treatment. There is no mention however if those patients resist or not treatment! They could not verify PKR activation because of the lack of suitable antibodies. Why they did not check the level of the downstream effectors ATF4 and p21.

REPLY: The tumors were obtained from patients that developed resistance to Trastuzumab. We clarify it in the Results section of revised manuscript and legend of Suppl. Fig. 6.

We ran quality control experiments with different commercially available ATF4 antibodies and found them unsuitable for the staining of the breast tumor samples (high background of unspecific staining). Our quality control data are at the disposal of the Reviewers and Journal.

We have assessed P21 levels in the HER2+ breast tumors from patients that developed resistance to Trastuzumab. We found that P21 expression did not significantly differ between tumor samples of the same patient before and after Trastuzumab therapy (see below). This result agrees to a previous study by Alexander et al. J. Biol. Chem. 292:748-759 (2017) showing that P21 levels are induced in breast cancer patients that respond to Trastuzumab but in those patients that develop resistance to Trastuzumab. This is also supported by our findings showing that P21 is upregulated by Trastuzumab in the sensitive but not resistant BT474 cells (Suppl. Fig. 5b).

They addressed the prognostic value of PeIF2 α from the analyses of TMAs derived from patients with HER2+ metastatic breast cancer. Low PeIF2 α was independent of HER2 expression whereas high PeIF2 α seems to correlate with increased HER2 levels. Since PeIF2 α is known to modulate the proteome, it will be helpful to address if the high level of PeIF2 α could affect HER2 translation.

REPLY: Increased levels of phosphorylated eIF2 α by SAL003 did not exhibit an effect on HER2 expression levels as shown below (Fig. 5a in the revised version).

Although not mentioned, I expect that the data that are generated from these analyses rely on the use of a single anti-PeIF2a antibody. The use of a second anti-PeIF2a antibody or antibodies specific to downstream targets is recommended to confirm these important patient studies.

REPLY: The antibody for the staining of phosphorylated eIF2α in our study is from Abcam (Cat#: ab32157). We have compared the Abcam antibody to an antibody from Cell Signaling Technology (CST Cat# 3597), which is suitable for immunohistochemistry as per manufacturer's recommendation. Staining of identical human breast tumor sections with the two antibodies resulted in the same pattern. However, the Abcam antibody was of superior quality. Previous work from our lab demonstrated the specificity of the Abcam antibody for phosphorylated eIF2α in human tumor cells grown in nude mice (Aging 2013;5:884-901). The antibody was previously distributed by Novus and is now sold by Abcam. The data comparing the two different antibodies is now included in the revised manuscript as Suppl. Fig. 8.

Reviewer #2

Figure 7 uses an *in vivo* HER2+ gastric cancer PDX model to essentially validate *in vitro* results using breast cancer cells in Fig. 6. Switching to a gastric model for the last figure is confusing and not congruent with rest of manuscript (or title of manuscript). Authors should perform *in vivo* experiments with BT474-R cells with these drugs or at minimum show that TZ & SAL regulated pathway (as shown in Fig. 6A) also exists in gastric cancer PDX model (and remove “breast” from title of manuscript).

REPLY: The Trastuzumab-resistant BT474 cells can form tumors in immune deficient mice but after prolonged time (4-5 months) [e.g. PLOS ONE 7; e47995 (2012); PLOS ONE 8 :e70641 (2013)]. Instead, we used the HER2+ gastric PDX, which forms tumors in NOG mice faster (Fig. 6a). We have analyzed the activation of the eIF2 α -P pathway in 4 tumor samples from mice implanted with HER2 gastric PDX and treated with SAL003 and/or Trastuzumab (see below). The data show the activation of eIF2 α -P and downstream anti-tumor pathways by the treatments. We have included these data in the revised manuscript (now in Suppl. Fig. 7 plus quantifications). We have modified the title of the manuscript as recommended.

Having the clinical data in middle of manuscript is confusing and made manuscript difficult to read. It may be better to have it at end of manuscript instead of going back and forth from animal/signaling data (same for abstract).

REPLY: We have modified the presentation of the data in the revised manuscript as recommended.

Authors should discuss the discrepancy from clinical data and experimental models. Cell models show that HER2 overexpression reduces phospho- eIF2 α and treatment with TZ increases phospho-eIF2 α . However, authors found high phospho- eIF2 α correlates with high HER2 levels.

REPLY: Please see also our response to Reviewer 1. SAL003 does not alter HER2 expression in BT474 cells (new data in Fig. 5a) suggesting that phosphorylated eIF2 α does not impact on HER2 expression directly. We do not know why increased eIF2 α phosphorylation correlates with increased HER2 expression in the clinical samples. We also found that the development of resistance to Trastuzumab is associated with increased HER2 expression in BT474 cells (Suppl. Fig. 5b). Perhaps pathways that act in parallel with phosphorylated eIF2 α impact on HER2 expression in breast tumors from treated patients. This is now stated in the Discussion of the revised manuscript.

Figure 2C: Do these tumors also have reduced KI-67 and increased apoptosis?

REPLY: The new data is shown below and in Suppl. Fig. 3 of the revised manuscript.

Figure 3 should be part of Figure 2 (Its in the same section of Results).

REPLY: We have included the quantifications in Fig.2 and Fig.3 as well as new data in Fig. 3 (growth of proficient and ATF4-deficient tumor cells). For the clarity of the presentation, we have kept the 2 figures separately.

Fig. 4D. Should show effects of JNK inhibitor on p-JNK in cells (not clear how authors know drug is working).

REPLY: The pharmacological inhibition of JNK was verified by the immunoblotting of phosphorylated JNK as shown below. Data is now shown in Suppl. Fig. 4.

4. Stats missing from Fig. 7A.

REPLY: Statistical analysis of the data is now included in Fig. 6a (previous Fig. 7a) and all figures.

5. Model of Fig. 7D is confusing. Looks like HER2 is activating PKR-P (when its really inhibiting it). A better model is needed.

REPLY: The model is corrected (now in Fig. 6d).

REVIEWERS' COMMENTS:

Reviewer #1 (Remarks to the Author); expert in breast cancer resistance:

The authors nicely addressed the majority of my previous comments.

The reviewer has an issue with the presented model. This is not a new comment as it is related to my previous point concerning the role of ATF4-p21 in the resistance to Trastuzumab treatment. As presented, the proposed model, though not explicitly mentioned, suggests that chemoresistance occurs through ATF4 and p21, downstream of $PeIF2\alpha$. Most of the data that are shown in this manuscript indicated a role of ATF4-p21 in promoting mouse tumor formation and/or growth. No functional data demonstrating the role of ATF4 and p21 per se in preventing the resistance of human cancer cells/tumors to Trastuzumab treatment is provided. Therefore, if the authors want to keep their model in its present form, new data assessing directly the potential role of ATF4 and p21 in human cancer cells or tumors in the prevention of Trastuzumab resistance are then required. This is important considering the conflictual roles of both ATF4 and p21 in cancer. Whatever the downstream $PeIF2\alpha$ pathways, the key finding here is that high $PeIF2\alpha$ level may be used as a surrogate biomarker of the outcome of treatment and that hyper-activating $PeIF2\alpha$ may prove beneficial for patient treatment. Whether the therapeutic effects of $PeIF2\alpha$ in sensitizing tumors to Trastuzumab treatment is due to the activation of the downstream ATF4-p21 arm is however not addressed in this manuscript. Because this question is not the focus of the present study, we believe that the revised manuscript, with a slight modification of the model, is suitable for publication in Nature Comm.

Reviewer #2 (Remarks to the Author); expert in breast cancer models and signalling:

Authors have addressed all major issues. Manuscript is now acceptable for publication.

Reviewer #1:

The authors nicely addressed the majority of my previous comments. The reviewer has an issue with the presented model. This is not a new comment as it is related to my previous point concerning the role of ATF4-p21 in the resistance to Trastuzumab treatment. As presented, the proposed model, though not explicitly mentioned, suggests that chemoresistance occurs through ATF4 and p21, downstream of PeIF2 α . Most of the data that are shown in this manuscript indicated a role of ATF4-p21 in promoting mouse tumor formation and/or growth. No functional data demonstrating the role of ATF4 and p21 per se in preventing the resistance of human cancer cells/tumors to Trastuzumab treatment is provided.

Therefore, if the authors want to keep their model in its present form, new data assessing directly the potential role of ATF4 and p21 in human cancer cells or tumors in the prevention of Trastuzumab resistance are then required. This is important considering the conflictual roles of both ATF4 and p21 in cancer. Whatever the downstream PeIF2 α pathways, the key finding here is that high PeIF2 α level may be used as a surrogate biomarker of the outcome of treatment and that hyper-activating PeIF2 α may prove beneficial for patient treatment. Whether the therapeutic effects of PeIF2 α in sensitizing tumors to Trastuzumab treatment is due to the activation of the downstream ATF4-p21 arm is however not addressed in this manuscript. Because this question is not the focus of the present study, we believe that the revised manuscript, with a slight modification of the model, is suitable for publication in Nature Comm.

REPLY: We thank the Reviewer for the constructive comments. We agree that the link between ATF4 and the downstream anti-tumor pathways, namely P21^{CIP1} and JNK, as much as the chemoresistance to Trastuzumab is concerned, has not been addressed in our study. Specifically, it is currently unclear whether the ATF4-dependent anti-tumor pathways, which were characterized in mouse NEU breast tumor cells, are enough to mediate the anti-tumor effects of Trastuzumab in single or combined treatments with the eIF2 α -phosphatase inhibitor SAL003. As such, **we have revised our model in Fig. 6d accordingly** and have clarified this point in the Discussion (page 13, last 2 sentences of the first paragraph). Moreover, we have included a schematic model in Fig. 4e, which summarizes the signaling properties of the anti-tumor PKR/eIF2 α -P arm from the analyses of the mouse NEU breast tumor cells (Figs. 1-4), which we believe will be helpful to the readers.